# A prevalent and culturable microbiota links ecological balance to clinical stability of the human lung after transplantation

Sudip Das [1,5], Eric Bernasconi [2,5✉], Angela Koutsokera[2], Daniel-Adrien Wurlod[2], Vishwachi Tripathi[1,4], Germán Bonilla-Rosso [1], John-David Aubert [2], Marie-France Derkenne[2], Louis Mercier[2], Céline Pattaroni[2,3], Alexis Rapin [2], Christophe von Garnier[2], Benjamin J. Marsland [2,3,6], Philipp Engel [1,6✉] & Laurent P. Nicod [2,6]

There is accumulating evidence that the lower airway microbiota impacts lung health. However, the link between microbial community composition and lung homeostasis remains elusive. We combine amplicon sequencing and bacterial culturing to characterize the viable bacterial community in 234 longitudinal bronchoalveolar lavage samples from 64 lung transplant recipients and establish links to viral loads, host gene expression, lung function, and transplant health. We find that the lung microbiota post-transplant can be categorized into four distinct compositional states, 'pneumotypes'. The predominant 'balanced' pneumotype is characterized by a diverse bacterial community with moderate viral loads, and host gene expression profiles suggesting immune tolerance. The other three pneumotypes are characterized by being either microbiota-depleted, or dominated by potential pathogens, and are linked to increased immune activity, lower respiratory function, and increased risks of infection and rejection. Collectively, our findings establish a link between the lung microbial ecosystem, human lung function, and clinical stability post-transplant.

[1] Department of Fundamental Microbiology, Biophore, University of Lausanne, Lausanne, Switzerland. [2] Department of Respiratory Medicine, Lausanne University Hospital and University of Lausanne, Lausanne, Switzerland. [3] Department of Immunology and Pathology, Central Clinical School, Monash University, Melbourne, Victoria, Australia. [4] Present address: Biozentrum, University of Basel, Basel, Switzerland. [5] These authors contributed equally: Sudip Das, Eric Bernasconi.. [6] These authors jointly supervised this work: Philipp Engel, Benjamin J. Marsland, Laurent P. Nicod. ✉ email: eric.bernasconi@chuv.ch; philipp.engel@unil.ch

Recent studies have shown that diverse bacterial communities are present in the lower respiratory tract of healthy humans[1–6]. These communities are predominated by the same phyla as the oral and gastrointestinal microbiota (*Bacteroidetes*, *Firmicutes*, *Actinobacteria*, *Proteobacteria*). However, their phylogenetic composition, total bacterial load, and temporal-spatial dynamics are distinct owing to the characteristic physicochemical, anatomical, and immunological conditions of the lung, which makes this organ a distinct microbial habitat with specific host-microbe interactions[7,8].

Several independent studies have shown that supraglottic taxa (i.e., bacteria found in the human oropharyngeal area) such as *Streptococcus*, *Prevotella*, and *Veillonella* are major constituents of the healthy lower respiratory tract microbiota. These bacteria have been proposed to contribute to the immunological development and homeostasis of the human lung, as their presence correlates with an increased pro-inflammatory response during postnatal immune maturation as well and lung function in adulthood[4,9]. Shifts in microbial community composition, together with a decrease in bacterial diversity[10,11], have been associated with various respiratory diseases such as Chronic Obstructive Pulmonary Disease (COPD), Idiopathic Pumonary Fibrosis and asthma. Together, these findings suggest that the lower respiratory microbiota is linked to the health state of the human lung and hence may play important roles for maintaining lung homeostasis.

Formidable challenges are associated with studying the lower respiratory tract microbiota. Firstly, the sampling of the human lung, which is best achieved by collecting bronchoalveolar lavage fluid (BALF) during bronchoscopy[12], is an invasive procedure which implies that it is rarely performed in healthy individuals. Consequently, large datasets from healthy individuals—including longitudinal studies that would inform about the dynamics of the human lung microbiota—are scarce. Secondly, the relatively low bacterial biomass in the human lung increases the risk of describing contaminants as being part of the respiratory tract microbiota. This can skew diversity measures of the lower respiratory tract microbiota, in particular when solely relying on relative abundance data[5]. Thirdly, while several studies have isolated viable bacteria from lung samples[6,13,14], a bacterial collection that can serve as a public resource has not been established, and little is known about the physiology and growth characteristics of lung isolates. Therefore, our current understanding of the ecological properties of different lung microbiota members and how these are linked to the environmental conditions in the lung ecosystem (such as immune state) remains limited.

Studying the microbiota in the context of lung transplantation can provide important insights about the crosstalk between the respiratory tract microbiota and the host[15]. Lung transplant recipients undergo post-transplant follow-up, in which BALF is collected to monitor the health state of the transplanted organ. This offers unique opportunities for longitudinal studies on the lung microbiota composition and allows establishing links to the host's immune state and to clinical metadata. Due to different types of clinical complications such as infection[16], acute cellular or humoral rejection[17] and Chronic Lung Allograft Dysfunction (CLAD)[18], the transplanted lung also offers the opportunity to study the respiratory microbiota[19–22] under a wide range of ecological conditions. A better understanding of the dynamics of the lung ecosystem in this context can ultimately help limit the burden of morbidity and mortality associated with post-transplant complications and promote graft survival.

Recent studies on lung transplants have provided insights about the distribution of the microbiota along the conducting and respiratory airways[23], or the adaptation of opportunistic pathogens to the lung environment[24]. Moreover, there is accumulating evidence that the immune state of the transplanted lung correlates with changes in the composition of the lung microbiota[20,25]. High abundance of opportunistic pathogens such as members of the genera *Staphylococcus* and *Pseudomonas* have been linked to pro-inflammatory responses in the transplanted lung[25,26], and were also found in respiratory diseases such as COPD and asthma[27,28]. These bacteria activate macrophages and induce a strong inflammatory response after transplantation, reflected by high levels of tumor necrosis factor-α and cyclooxygenase-2[25]. This is in contrast to certain strains of non-pathogenic *Streptococcus pneumoniae*, whose abundance has been linked to low inflammation and tissue repair and remodeling[25]. Sustained inflammatory reactions and uncontrolled tissue remodeling can eventually lead to irreversible decline in lung function[29,30]. Collectively, these previous data suggest that the lung microbiota post-transplant can constitute different compositional states that may be linked to allograft function. However, quantitative analyses of these microbiota profiles are currently lacking, including the phylogenetic and physiological characterization of viable community members, and the links to the lung ecological environment and the clinical outcome post-transplant.

In this study, we characterized the airway microbiota in 234 longitudinal BALF samples from 64 lung transplant recipients. We combined culture-independent and culture-dependent analyses to identify the most prevalent lung bacteria post-transplant and to establish a collection of primary lung bacterial isolates. We linked the identified compositional changes in lung microbiota to host gene expression profiles, anellovirus loads and patient metadata to understand the importance of the ecological environment of the transplanted lung on clinical outcomes. Our findings show that BALF samples can be classified into four distinct compositional states (i.e., pneumotypes) similar to the enterotypes identified in the human gut[31]. These pneumotypes are distinguished by different community characteristics and distinct physiological properties of their predominant members. We show that pneumotypes are differentially associated with anellovirus loads, respiratory function, and both local and peripheral host immune responses, including those linked to allograft rejection. Taken together, our findings not only illustrate the strong links between lung health and local microbiota composition, but pinpoint underlying community characteristics and lung environmental conditions as well as provide a large resource of cultured isolates for future experimental approaches.

## Results

**Combined culture-dependent and culture-independent approach identifies the prevalent and viable bacterial community members of the human lung post-transplant.** To characterize the bacterial community composition of the lung microbiota post-transplant, we performed 16S rRNA gene amplicon sequencing of 234 longitudinal BALF samples from 64 lung transplant recipients collected over a 49-month period (Fig. 1a, Supplementary Table 1). A total of 7164 operational taxonomic units (OTUs) were identified, excluding OTUs contributing to reads in 11 negative control samples[32] (see "Methods", Supplementary Fig. 1a, Supplementary Data 1 and 2). In accordance with previous studies on BALF samples from healthy non-transplant individuals[4–6,26], we found that *Bacteroidetes* and *Firmicutes* followed by *Proteobacteria* and *Actinobacteria* are the most abundant phyla in the post-transplant lung (Fig. 1b). Prevalence analysis across all BALF samples showed that the community composition is highly variable with only 22 OTUs shared by ≥50% of the samples (Supplementary Fig. 1b, Supplementary Data 3). However, these 22 OTUs constituted 42% of the total

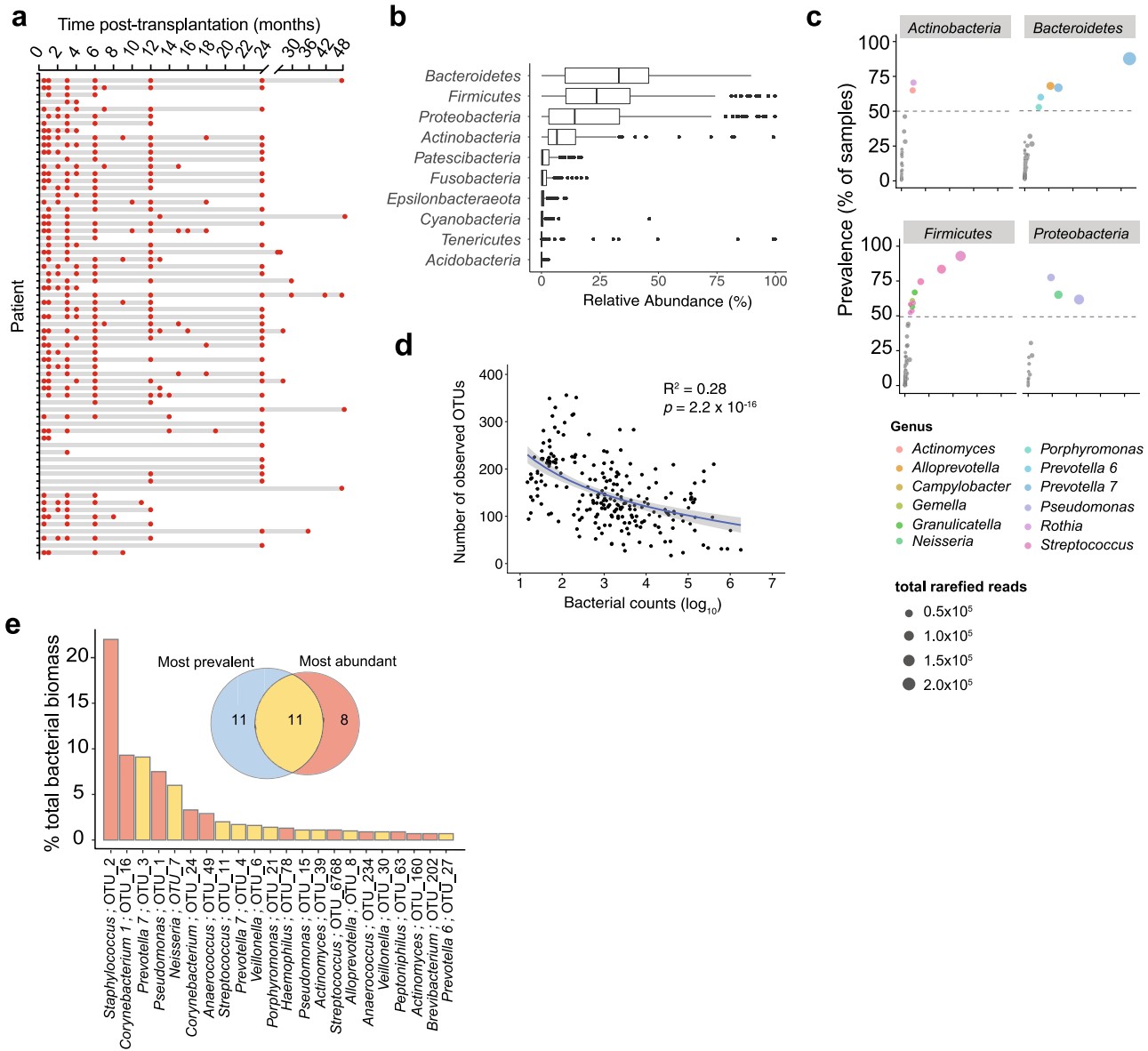

**Fig. 1 Combining BALF amplicon sequencing and bacterial culturing to deduce the microbial ecology of deep lung microbiota. a** Schematic of the sampling of Bronchoalveolar lavage fluid (BALF) from lung transplant recipients over time (months post-transplant). **b** Relative abundances (%) of most abundant phyla across BALF samples. Box plots show median (middle line), 25th, 75th percentile (box) and 5th and 95th percentile (whiskers) as well as outliers (single points). **c** Prevalence (% samples) vs contribution to total reads across samples for most abundant phyla. Dot color shows different genera and size show total rarefied reads. Gray dashed horizontal line shows prevalence ≥50%. **d** Scatter plot shows correlation between number of observed OTUs and bacterial counts per BALF sample obtained by quantifying 16S rRNA gene copies with qPCR. Linear regression is shown by the blue line with gray shaded area showing 95% confidence interval ($n = 234$, two-sided, $F(1, 232) = 91.04$, $P = 2.2 \times 10^{-16}$), Coefficient of correlation; $R^2 = 0.28$. **e** Bar chart shows lung taxa (genera; OTU IDs) that contributed ≥75% of total bacterial biomass across samples ($n = 234$). Venn diagram inset shows overlap (yellow) between the most prevalent (≥50% incidence, light blue) and the most abundant (≥75% total count, red) taxa in the transplanted lung. Bar colors also show the same.

number of rarefied reads, indicating that they are predominant members of the post-transplant lung microbiota (Fig. 1c, Supplementary Fig. 1c, Supplementary Table 2, Supplementary Data 3). They belonged to the genera *Prevotella 7*, *Streptococcus*, *Veillonella*, *Neisseria*, *Alloprevotella*, *Pseudomonas*, *Gemella*, *Granulicatella*, *Campylobacter*, *Porphyromonas* and *Rothia*, the majority of which are also prevailing community members in the healthy human lung[3,5,7,26], suggesting a considerable overlap in the overall composition of the lung microbiota between the healthy and the transplanted lung.

Differences in bacterial loads between samples can skew community analyses when based on relative abundance profiling

alone. Therefore, we used qPCR to determine the total copies of the 16S rRNA gene as an estimate for bacterial counts, and normalized the abundances of each OTU across the 234 samples (absolute abundance). We found that the bacterial counts vastly differed between samples, ranging between $10^1$ and $10^6$ gene copies per ml of BALF (Supplementary Fig. 1d). The number of observed OTUs increased with decreasing counts (Fig. 1d) suggesting that a large fraction of the OTUs were detected in samples of low bacterial biomass and hence represent either transient or extremely low-abundant community members, or sequencing artefacts and contaminations. In turn, 19 of the 7164 OTUs constituted >75% of the total bacterial biomass detected

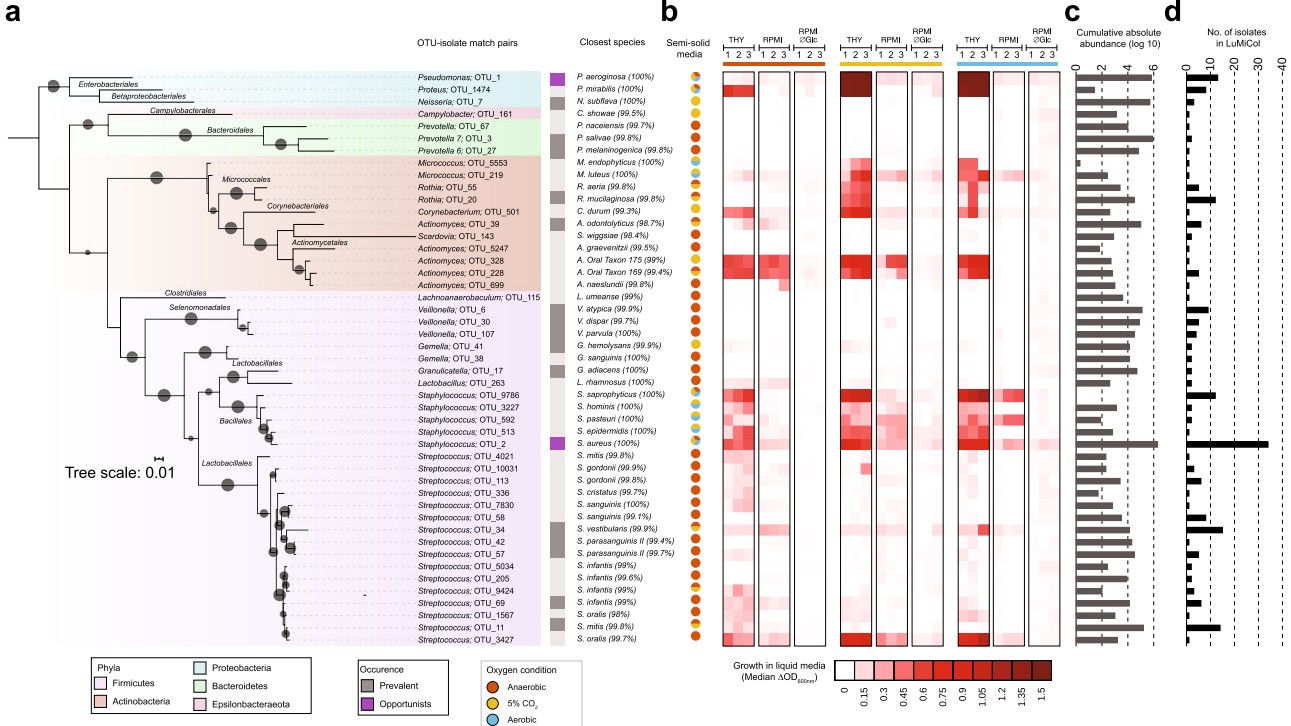

**Fig. 2 A lung microbiota culture collection (LuMiCol) reveals extended diversity and phenotypic characteristics of the lower airway bacterial community. a** Phylogenetic tree of the 47 OTU-isolate matching pairs inferred with FastTree. Branch bootstrap support values (size of dark gray circles) ≥80% are displayed. **b** Growth characteristics of each OTU-isolate matching pair in three different oxygen conditions (Anaerobic - light brown, 5% $CO_2$-yellow, aerobic-light blue, $n = 3$). Column with pie charts shows growth on semi-solid agar. Heatmap shows median change in Optical Density (OD) at 600 nm growth in three different liquid media (THY, RPMI, RPMI without glucose) over 3 days. **c** Cumulative counts of each OTU-isolate matching pair across all BALF samples (gray). **d** Number of isolates in Lumicol (black) per OTU-isolate matching pair. Taxa are labeled as genus; OTU ID, with an indication of whether they are prevalent (gray rectangle) or opportunistic (magenta rectangle) in the lower airway community. The names of the closest hit in databases: eHOMD and SILVA are used as species descriptor.

across the 234 BALF samples (Fig. 1e). This included 11 of the 22 most prevalent OTUs (see above) plus eight OTUs that were detected in only a few samples but at very high abundance (*Staphylococcus;* OTU_2, *Corynebacterium 1;* OTU_16 *and* OTU_24, *Anaerococcus;* OTU_49 and OTU_234, *Haemophilus;* OTU_78, *Streptococcus;* OTU_6768, *Peptoniphilus;* OTU_63, Supplementary Table 2). It is important to differentiate these opportunistic colonizers from other community members with low incidence, as they reached very high bacterial counts in some samples with potential implications for lung health.

To demonstrate the viability of prevalent lung microbiota members and to establish a reference catalogue of bacterial isolates from the human lung for experimental studies, we complemented the amplicon sequencing with a bacterial culturing approach (Supplementary Fig. 2). We cultivated 21 random BALF samples from 18 individuals, on 15 different semi-solid media (both general and selective) in combination with 3 oxygen concentrations; aerobic, 5% $CO_2$, and anaerobic (See "Methods" and Supplementary Table 3), representing 26 different conditions. We cultured fresh BALF immediately upon extraction (within 2 h), as we observed loss in bacterial diversity upon cultivating frozen samples. This resulted in a total of 300 bacterial isolates, representing 5 phyla, 7 classes, 13 orders, and 17 families from which we built an open-access biobank called the **Lu**ng **Mi**crobiota culture **Col**lection (LuMiCol, Supplementary Data 4, https://github.com/sudu87/Microbial-ecology-of-the-transplanted-human-lung).

To examine the extent of overlap between bacteria in LuMiCol and the diversity obtained by amplicon sequencing, we included 16S rRNA gene sequences from 215 isolates that passed our quality filter into the community analysis, which allowed for the

retrieval of OTU-isolate matching pairs[32] (Methods). We found that 213 isolates matched to 47 OTUs (Fig. 2a, c, Supplementary Data 5), including 17 of the most prevalent and abundant bacteria (Fig. 1e, Supplementary Table 2). As expected, bacteria with high abundance in the amplicon sequencing-based community analysis were isolated more frequently, with *Firmicutes* revealing the highest isolate diversity (Fig. 2a–c, Supplementary Data 4, 5) and being recovered under the most diverse culturing conditions.

In summary, our results from the combined culture-dependent and culture-independent approach show that the lung microbiota post-transplant is highly variable in terms of both bacterial load and community composition with many transient and low-abundant bacterial taxa. However, a few community members display relatively high prevalence and/or abundance suggesting that they represent important colonizers of the human lung.

**LuMiCol informs on the diversity and metabolic preferences of culturable human lung bacteria**. We characterized the culturable community members of the lower respiratory tract contained in LuMiCol by testing a wide range of growth conditions and phenotypic properties (see "Methods"). The majority of the cultured isolates could taxonomically be assigned at the species level based on genotyping of the 16S rRNA gene V1-V5 region. However, the limited taxonomic resolution offered by this method does not allow to discriminate between closely related strains, which can include both pathogenic and non-pathogenic bacteria. Hence for *Streptococcus*, we additionally tested for type of hemolysis (alpha, beta, or gamma) and resistance to optochin, which differentiates the pathogenic pneumococcus and the non-

pathogenic viridans groups (Fig. 2a, Supplementary Fig. 2b, c). This demonstrated that the 16 *Streptococcus* OTU-isolate pairs belong to the viridans group of streptococci (VS)[33]. Interestingly, these isolates exhibited the highest genotypic and phenotypic diversity throughout our collection and belonged to five OTUs among the 22 most prevalent community members, with *Streptococcus mitis* (OTU_11) present in 93.6% of all samples.

BALF from healthy individuals contains amino acids, citrate, urate, fatty acids, and antioxidants such as glutathione but no detectable glucose[34], which is associated with increased bacterial load and infection[35–37]. To get insights into basic bacterial metabolism, we assessed the growth of all 47 isolates matching an OTU under different oxygen concentrations. We used undefined rich media (Todd-Hewitt Yeast extract) and defined low-complexity liquid media (RPMI 1640), including a glucose-free version to mimic the deep lung environment (see "Methods"). Despite the presence of oxygen in the human lung, the majority of the isolates were either obligate or facultative anaerobes (Fig. 2a), including some of the most prevalent members (*Prevotella melaninogenica* (OTU_3), *Streptococcus mitis* (OTU_11), *Veillonella atypica* (OTU_6) and *Granulicatella adiacens* (OTU_17). A similar trend was also observed in liquid media under anaerobic conditions, with the exception of the genera *Prevotella*, *Veillonella* and *Granulicatella*. Most streptococci from the human lung grew best in complex liquid media containing glucose under anaerobic conditions, including the most prevalent species in our cohort, *S. mitis* (OTU_11) (Fig. 2b). However, noticeable exceptions were *S. vestibularis* (OTU_34), *S. oralis* (OTU_3427 and OTU_1567), and *S. gordonii* (OTU_10031), which grew equally well in the presence of oxygen and in low-complexity liquid medium (Fig. 2b). Most *Actinobacteria* grew best on rich medium in the presence of 5% $CO_2$, with an exception of *Actinomyces odontolyticus* (OTU_39), which required anaerobic conditions. Some *Actinobacteria* grew equally well in anaerobic conditions as in the presence of 5% $CO_2$, i.e., *Corynebacterium durum* (OTU_501), *Actinobacteria* sp. oral taxon (OTU_328 and OTU_228).

The two most predominant opportunistic pathogens in our lung cohort, *P. aeruginosa* (OTU_1) and *S. aureus* (OTU_2), grew best in rich liquid medium in the presence of oxygen (Fig. 2c), although these also grew to lower degree under anaerobic conditions. These results indicate that changes in the physicochemical conditions in the lung may favor the growth of these two opportunistic pathogens. In summary, our observations from the bacterial culture collection provide first insights into the phenotypic properties of human lung bacteria and will serve as a basis for future experimental work.

**Identification of four compositionally distinct pneumotypes post-transplant using machine learning based on ecological metrics.** To detect and characterize differences in bacterial community composition between BALF samples from transplant patients, we clustered the samples using an unsupervised machine learning algorithm based on pairwise Bray–Curtis dissimilarity[32] (beta diversity, See "Methods", Supplementary Data 6). This segregated the samples into four partitions around medoids (PAMs) at both phylum and OTU level (Fig. 3a, b, Supplementary Fig. 3a, b). We refer to these clusters as "pneumotypes" PAM1, PAM2, PAM3, and PAM4 (Supplementary Table 4). PAM1 formed the largest cluster consisting of the majority of samples ($n = 115$) followed by PAM3 ($n = 76$), PAM2 ($n = 19$), and PAM4 ($n = 24$) (Supplementary Data 7). Examination of various diversity measures (Species occurrence, OTU diversity, OTU richness, Fig. 3c–e), distribution of the dominant community members (Fig. 3f), and bacterial counts (16S rRNA gene

copies, Fig. 3g) revealed distinctive characteristics between the four pneumotypes.

PAM1 showed the highest similarity in community composition between samples (Species occurrence/Sorenson's Index, Fig. 3c), and had intermediate levels of diversity and bacterial load (Fig. 3d, e and g, Supplementary Fig. 3c). Twenty of the 22 most prevalent community members were enriched in incidence and abundance in PAM1 when compared to the other PAMs (ART-ANOVA, FDR, abundance $P < 0.01$, Fig. 3h, Supplementary Table 5) with five OTUs occurring in >90% of the samples (incidence); *P. melaninogenica* (OTU_3, 97.4%), *S. mitis* (OTU_11, 99.1%), *V. atypica* (OTU_6, 93.9%), *V. dispar* (OTU_30, 93%) and *G. adiacens* (OTU_17, 93%). Contrastingly, two OTUs (*P. aeruginosa*; OTU_1 and *P. fluorescens*; OTU_15) had neither a higher incidence nor a higher abundance in PAM1 (Fig. 3h, Supplementary Table 5). Thus, PAM1 samples harbor balanced bacterial communities of relatively high similarity composed of the most prevalent bacteria across our data set. Henceforth, we refer to this PAM as the 'balanced pneumotype' (Pneumotype$_{Balanced}$).

In contrast to Pneumotype$_{Balanced}$, PAM2 and PAM4 harbored lower bacterial diversity (Fig. 3d) and OTU richness (Fig. 3e), were dominated by a single community member (Fig. 3f), and had higher bacterial loads (Fig. 3g). In these two PAMs, the taxa associated with Pneumotype$_{Balanced}$ had a low sample incidence and absolute abundance compared to the other PAMs (Supplementary Fig. 4a, b). *S. aureus* (OTU_2), *Corynebacterium* (OTU_24) and *Anaerococcus* (OTU_49) were enriched in PAM2 (ART-ANOVA, FDR, $P < 0.001$, Supplementary Fig. 4a), while *Haemophilus* (OTU_78) and *P. aeruginosa* (OTU_1 & OTU_15) dominated PAM4 (ART-ANOVA, FDR, $P < 0.001$, Supplementary Fig. 4b). We refer to these as 'Pneumotype$_{Staphylococcus}$' (PAM2) and 'Pneumotype$_{Pseudomonas}$' (PAM4), with the major species known to be potential pathogens that proliferate rapidly in lung, under a variety of pathological respiratory conditions[38,39], in line with the high loads observed in our setting (Fig. 3g).

The fourth cluster identified, PAM3, exhibited the lowest between-sample similarity in species composition (Fig. 3c), the highest OTU diversity and richness (Fig. 3d, e), and lowest dominance (Fig. 3f). The samples in this PAM were characterized by remarkably low bacterial loads, up to two orders of magnitude below samples in other PAMs (Fig. 3g, Supplementary Fig. 3c), suggesting a depauperated microbiota that has been associated with dysbiotic physiological states of the gut microbiota[40]. Consequently, the high OTU richness detected in PAM3 samples is likely due to over-sequencing of rare or transient species, or sequencing artefacts. This is also supported by the fact that the 30 predominant microbiota members in PAM3 were significantly reduced in their incidence and abundance compared to the other PAMs (ART-ANOVA, FDR, $P < 0.001$, Fig. 3i). We refer to this PAM as the 'microbiota-depleted' pneumotype (Pneumotype$_{MD}$).

Qualitative culture results obtained from matched BALF and bronchial aspirate (BA) on selective media further reinforced the genuine existence of the four pneumotypes (Fig. 3j, Supplementary Fig. 5). BALF samples from Pneumotype$_{Balanced}$ had the highest percentage of matches to the oropharyngeal microbiota, which consists of many of the bacteria predominant in this pneumotype (e.g., *Streptococcus* or *Veillonella*). Similarly, culture results of BALF samples with Pneumotype$_{Staphylococcus}$ and Pneumotype$_{Pseudomonas}$ were most frequently positive for *S. aureus*/*Corynebacterium* spp. and *P. aeruginosa*, respectively, while those obtained for BALF samples with Pneumotype$_{MD}$ were often culture negative (Fig. 3j, Supplementary Fig. 5a). A similar picture was observed for BA (Fig. 3j, Supplementary Fig. 5b),

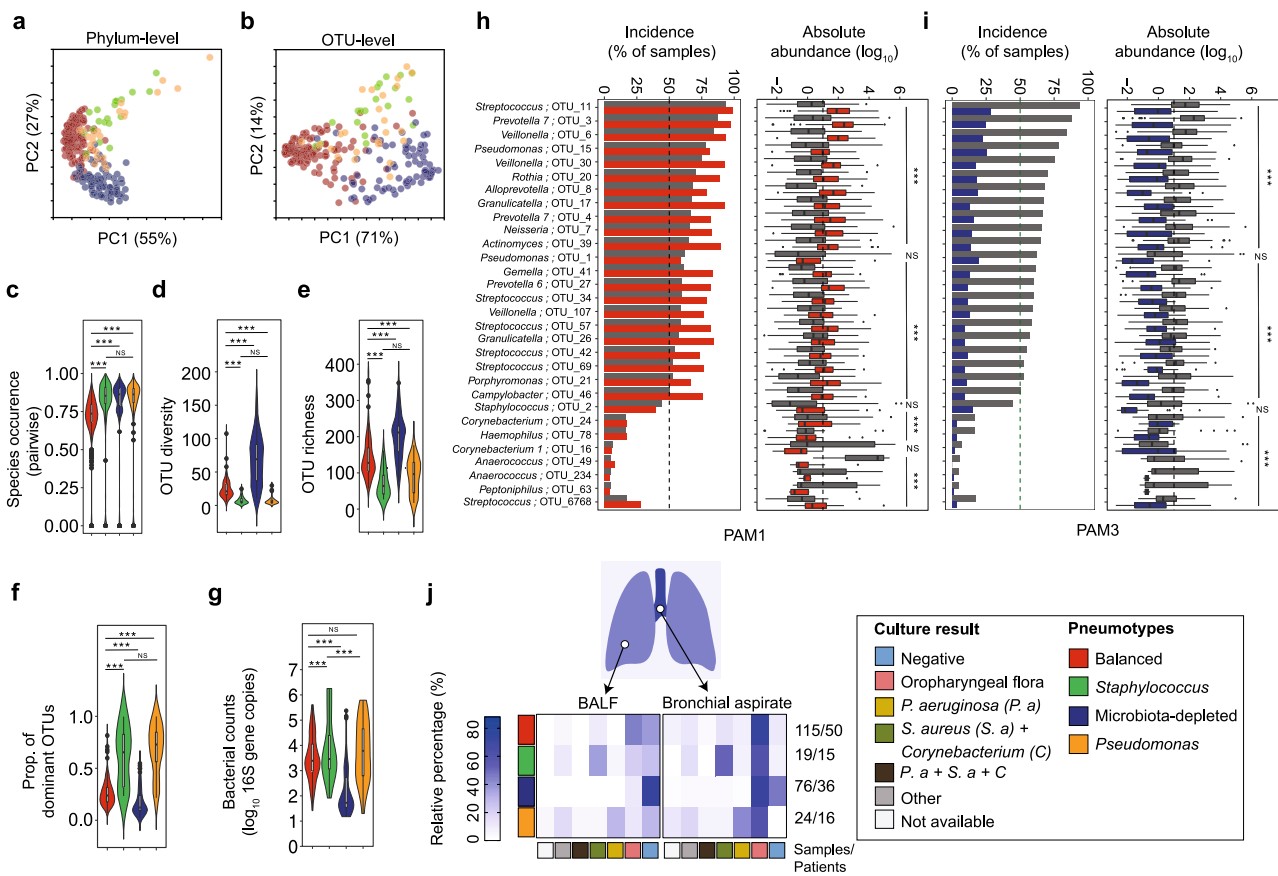

**Fig. 3 Bacterial communities of the lung post-transplant fall into four 'pneumotypes' with distinct community characteristics. a, b** Principal component analysis shows Partition around medoids (PAMs) at phylum and OTU level respectively generated by *k*-medoid-based unsupervised machine learning using Bray–Curtis dissimilarity (occurrence and abundance). Pneumotypes are color coded: Balanced (red, *n* = 115), *Staphylococcus* (green, *n* = 19), Microbiota-depleted (MD, blue, *n* = 76), and *Pseudomonas* (orange, *n* = 24). **c–g** Violin plots show distributions of pairwise species occurrence (Sorenson's index, PERMANOVA, two-sided, $F(3, 229) = 8.49$, $P = 9.9 \times 10^{-5}$), OTU diversity (Kruskal–Wallis test, $\chi^2 = 89.2$, df = 3, two-sided, $P = 2.2 \times 10^{-16}$), OTU richness (ANOVA, $F(3, 229) = 43.9$, two-sided, $P = 2.2 \times 10^{-16}$), proportion of most dominant OTUs (Kruskal–Wallis test, $\chi^2 = 94.45$, df = 3, two-sided, $P = 2.2 \times 10^{-16}$), and total bacterial counts (ANOVA, $F(3, 229) = 43.9$, two-sided, $P = 2.2 \times 10^{-16}$), respectively, across the four pneumotypes. **h, i** Enrichment analysis of prevalence (green dotted line ≥50%) and absolute abundance across all samples of the 30 most dominant taxa (i.e., OTUs) in Pneumotype_Balanced and Pneumotype_MD respectively, when each was compared to the other three combined pneumotypes (gray boxes). Differential abundances after enrichment analysis was calculated between each PAM and the other three PAMs combined, using ART-ANOVA. **j** Heatmap shows relative percentage of taxa (right colored panel) cultured from paired samples of Bronchial aspiration (BA) and Bronchoalveolar lavage fluid (BALF) from each pneumotype (left colored panel). Oropharyngeal flora mainly corresponds to Pneumotype_Balanced (i.e., *Streptococcus*, *Prevotella*, *Veillonella*). All box plots including insets show median (middle line), 25th, 75th percentile (box) and 5th and 95th percentile (whiskers) as well as outliers (single points). Multiple comparison of beta diversity indices was done by pairwise PERMANOVA (*adonis*) with False Discovery rate (FDR). Post hoc analyses (95% Confidence Interval) were done by using Tukey's test (ANOVA) or Dunn's test (Kruskal test) with False Discovery Rate (FDR) or least-squares means (ART-ANOVA) with False Discovery Rate (FDR). * $P < 0.05$, ** $P < 0.01$, *** $P < 0.001$, NS = not significant.

where however a higher percentage of positive cultures for oropharyngeal flora was observed compared to BALF, especially for Pneumotype_MD. This suggests differences in the microbiota between the two sample types, despite the known topographic continuity of microbial communities in the airways[1,3,7]. This is supported by the fact that for Pneumotype_Staphylococcus and Pneumotype_Pseudomonas, there was no association between pre- and post-transplant colonization by *S. aureus* ($\chi^2 = 0.047$, $P = 0.82$) and *P. aeruginosa* ($\chi^2 = 0.2$, $p = 0.65$) (Supplementary Fig 5c, d), respectively, in the upper respiratory tract.

Many of the OTUs of Pneumotype_MD could not be cultured in our bacterial culturing approach (Supplementary Table 3, 5, Supplementary Fig. 5), which together with the low bacterial abundance in corresponding samples, questions their relevance/ existence as lung microbiota members. In contrast, most of the major community members characteristic of the other three pneumotypes were represented by isolates in LuMiCol, including

the two opportunistic pathogens, *P. aeruginosa* (OTU_1) and *S. aureus* (OTU_2), providing the basis for future experimental work on the predominant species of these pneumotypes. Taken together, we identified four distinct bacterial communities in transplanted lung, which we refer to as pneumotypes, and validated them by culturing of BALF samples.

**Bacterial pneumotypes are linked to distinct host gene expression patterns.** The existence of bacterial pneumotypes with distinctive community composition suggests differences in the microenvironmental conditions of the human lung post-transplant, which could be echoed in other constituents of the lung ecosystem. We compared the median expression levels of 31 host genes belonging to seven functional categories across the four pneumotypes. These genes are involved in inflammation, immunoregulation, tissue remodeling and detection of bacteria

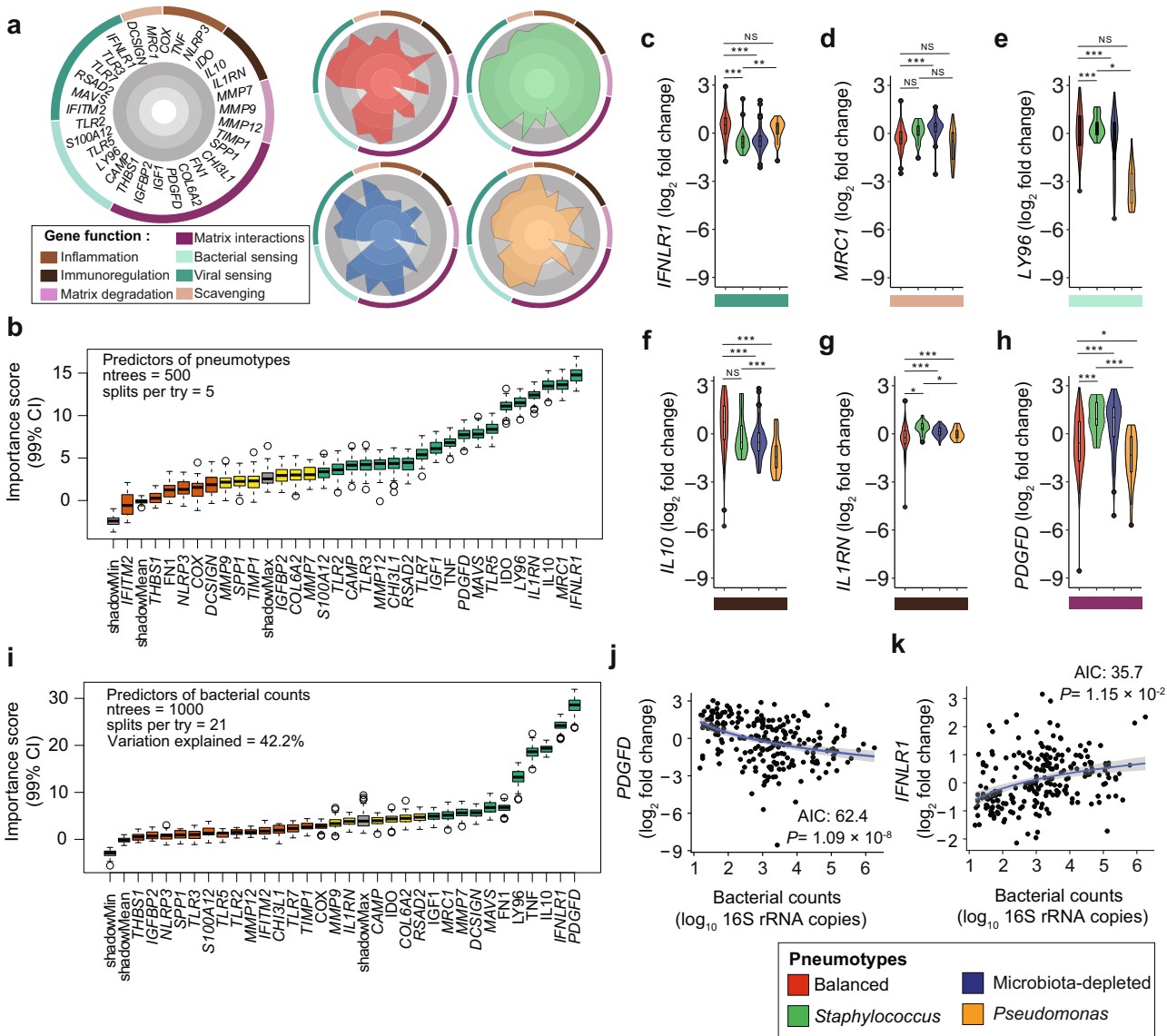

**Fig. 4 Host gene expression in the lung differs according to pneumotype and bacterial load. a** Radar plots show median-normalized expression of 31 host genes (radial axes) in the cell fraction of all BALF samples split by four pneumotypes. Circular distribution of genes is color-coded by seven functional categories and ticks (gray shading) show increase in expression from the inside to outside of the circle. **b** Importance scores (99% Confidence Interval) of host genes predictors of pneumotypes analyzed by Random Forest classification model and Boruta feature selection. Predictor genes are categorized into Confirmed (Green), Tentative (Yellow) and Rejected (Orange), ntrees = number of decision trees, splits per try = number of random predictors sub-sampled. **c–h** Violin plots showing distribution of expression ($\log_2$ fold) of host genes with Importance scores >10 for pneumotype prediction ($n = 229$): *IFNLR1* (Kruskal test, $\chi^2 = 59.18$, df = 3, two-sided, $P = 8.8 \times 10^{-13}$), *MRC1* (ANOVA, $F(3, 225) = 17.93$, two-sided, $P = 1.8 \times 10^{-10}$), *LY96* (Kruskal test, $\chi^2 = 33.87$, df = 3, two-sided, $P = 2.1 \times 10^{-7}$), *IL10* (Kruskal test, $\chi^2 = 58.82$, df = 3, two-sided, $P = 1.0 \times 10^{-12}$), *IL1RN* (Kruskal test, $\chi^2 = 58.02$, df = 3, two-sided, $P = 1.5 \times 10^{-12}$) and *PDGFD* (Kruskal test, $\chi^2 = 58.02$, df = 3, two-sided, $P = 8.7 \times 10^{-10}$) across the pneumotypes. **i** Importance scores (99% Confidence Interval) of host genes in predicting bacterial counts analyzed by Random Forest regression model and Boruta feature selection. In addition to ntrees and splits per try = number of random predictors sub-sampled at each step and percent variance explained. **j, k** Scatter plots show correlation between expression($\log_2$ fold) of *PDGFD* (stepwise regression AIC = 62.4, ANOVA, $t = -5.9$, $P = 1.09 \times 10^{-8}$) and *IFNLR1* (stepwise regression AIC:35.7, ANOVA, $t = 2.5$, $P = 1.15 \times 10^{-2}$) and bacterial counts ($\log_{10}$ 16 S rRNA copies) across samples ($n = 234$). Linear regression is shown by the blue line with gray shaded area showing 95% confidence interval. All box plots including insets show median (middle line), 25th, 75th percentile (box) and 5th and 95th percentile (whiskers) as well as outliers (single solid/hollow points). Post hoc analyses (95% Confidence Interval) were done by using Tukey's test (ANOVA) or Dunn's test (Kruskal test) with False Discovery Rate (FDR). * $P < 0.05$, ** $P < 0.01$, *** $P < 0.001$, NS = not significant.

and viruses (See "Methods", Fig. 4a), and their expression patterns differed across the four pneumotypes, with particularly high transcriptional activity in Pneumotype*Staphylococcus* (Fig. 4a). To identify the genes with the greatest power to discriminate between the four pneumotypes and between samples differing by bacterial counts, we applied a machine learning approach (Random Forest, See "Methods") based on the host gene expression in 234 BALF

samples. The Pneumotype*Balanced* was predicted with highest accuracy (92%), followed by Pneumotype*Pseudomonas* (83.4%) and Pneumotype*MD* (81.4 %), while no accuracy was achieved for Pneumotype*Staphylococcus* (Supplementary Table 5). We identified 6 of the 31 genes to have a particularly high predictive power *IFNLR1, MRC1, IL10, IL1RN, LY96, IDO* (Importance score >10; 99% Confidence Interval, Fig. 4b). *IFNLR1* encodes interferon

lambda receptor 1, which is involved in antiviral defence and epithelial barrier integrity[41]. This gene was up-regulated in samples with Pneumotype$_{Balanced}$ compared to the other three pneumotypes (Fig. 4c). MRC1 (Mannose Receptor C-Type 1)[42] and LY96 (Lymphocyte Antigen 96)[43] encode microbial poly-saccharide and lipopolysaccharide recognition proteins[43], respectively. Compared to samples with Pneumotype$_{Balanced}$, these two genes were up-regulated in Pneumotype$_{Staphylococcus}$ and Pneumotype$_{MD}$, and down-regulated in samples with Pneumotype$_{Pseudomonas}$ (Fig. 4d, e). Samples with Pneumotype$_{Balanced}$ further differed from those linked to the other three pneumotypes by higher expression of genes involved in immune modulation and peripheral immune tolerance (IL-10/Interleukin 10 and IDO1/Indoleamine 2,3-Dioxygenase 1, Fig. 4f, Supplementary Fig. 6), and a lower expression of IL1RN (Interleukin 1 Receptor Antagonist, Fig. 4g), produced as part of the inflammatory response to control the potentially deleterious effects of Interleukin-1 beta (IL-1β)[44].

Similarly, we found five genes with high discriminating power (Fig. 4i, importance score >10) for bacterial counts, of which two were particularly good predictors: PDGFD and IFNLR1. PDGFD encodes the D isoform of platelet-derived growth factor, which promotes the proliferation of cells of mesenchymal origin such as fibroblasts[45]. Expression of this gene was negatively correlated (Fig. 4j, AIC 62.4, $p < 0.001$) with bacterial abundance. In contrast, IFNLR1 expression positively correlated with bacterial abundance (Fig. 4k, AIC 35.7, $p < 0.05$). Accordingly, PDGFD expression was higher while IFNLR1 expression was lower in Pneumotype$_{MD}$ (Fig. 4c, h) as compared to the other pneumo-types, suggesting a link between the normal presence of bacteria in the lower respiratory tract and homeostatic levels of tissue remodeling, epithelial barrier integrity and host response to viruses. In summary, these results show that host-specific gene expression markers align with distinct bacterial states, high-lighting the existence of complex associations between different lung ecosystem characteristics.

**Anellovirus dynamics is associated with bacterial community and host physiology in lung.** The observed links between pneumo-types and antiviral defence prompted us to look into the tri-partite interactions between lung bacteria, viruses, and host. To this end, we quantified the load of the three genera of anelloviruses identified in humans (Alphatorquevirus, Betatorquevirus, and Gammatorquevirus) across the 234 BALF samples. In accordance with a previous study[46], we found that the transplanted lung con-tains high levels of anelloviruses, with Gammatorquevirus pre-dominating. Viral loads of the three genera peaked between 1.5 and 6 months after transplantation and decreased at later time points (Fig. 5a). Anellovirus load varied substantially between pneumo-types (Fig. 5b–d). All three viral genera were lowest in samples with Pneumotype$_{MD}$ and Alphatorquevirus showed particularly high levels in samples with Pneumotype$_{Pseudomonas}$ (Fig. 5b). Intra-individual pairwise analysis revealed a particularly strong decrease in load for longitudinal transitions from Pneumotype$_{Balanced}$ to Pneumotype$_{MD}$ and a corresponding increase for the inverse (Fig. 5e, Supplementary Fig. 7). We further identified four human genes: TLR3, IGF1, RSAD2, IFITM2, as important pre-dictors of anellovirus loads in BALF (Fig. 5f, See "Methods"). Of these, Toll-like Receptor 3 (TLR3) was positively correlated with total viral load (Fig. 5g, AIC 73.9, $p < 0.001$). This is consistent with the low viral load observed with Pneumotype$_{MD}$, where TLR3 was down-regulated (Fig. 5h). These findings link changes in the lung microbiota composition to changes in viral loads and host gene expression indicating possible implications for allograft outcome.

**Pneumotypes are linked to differential risk of post-transplant clinical complication.** A large set of clinical data (Supplementary Data 7) enabled us to associate differences in bacterial community composition, host gene expression, and anellovirus load to allo-graft and patient health status. Immunosuppression as well as prophylactic and therapeutic antibiotic usage were anticipated as major confounding factors. However, we found no association between the different pneumotypes and the main immunosup-pressive drugs (ANOVA, prednisone; $P = 7.68 \times 10^{-1}$, tacroli-mus; $P = 9.1 \times 10^{-1}$) used in our cohort, at the time of BALF sampling (Supplementary Fig. 8a, b). In contrast to what has been reported for blood plasma after transplantation[47], we also did not observe a correlation between anellovirus load and immunosup-pressive drug levels (Lm, prednisone; $P = 8.0 \times 10^{-1}$, tacrolimus; $P = 9.1 \times 10^{-1}$, Supplementary Fig. 8c, d, See "Methods"). How-ever, we observed a negative relationship between the number of antibiotics administered at the time of BALF sampling and the fraction of samples in Pneumotype$_{Balanced}$, and a positive rela-tionship with the fraction of Pneumotype$_{MD}$ samples (Fisher's test, $P = 2.0 \times 10^{-3}$, Fig. 6a). These observations thus suggest a link between intensive antibiotic use and a disturbance of the most balanced and compositionally stable lung microbiota profile.

We observed that a clinical diagnosis of infection was rare in the presence of Pneumotype$_{Balanced}$ and Pneumotype$_{MD}$, com-pared to Pneumotype$_{Staphylococcus}$ and Pneumotype$_{Pseudomonas}$ (Binomial linear model, Yes or No, $P < 0.001$ and $P = 0.016$, respectively; Fig. 6b). This confirms the results of our 16S rRNA gene analysis, which showed that Pneumotype$_{Staphylococcus}$ and Pneumotype$_{Pseudomonas}$ are dominated by opportunistic patho-gens, S. aureus and P. aeruginosa, respectively. It is also consistent with the finding of lower numbers of neutrophils (Fig. 6c), but not macrophages (Fig. 6d), in Pneumotype$_{Balanced}$ and Pneumo-type$_{MD}$ as compared to Pneumotype$_{Staphylococcus}$ and, to a lesser extent, Pneumotype$_{Pseudomonas}$, emphasizing that pneumotypes are associated with local conditions that differ in terms of recruitment of pro-inflammatory cells.

Lung transplant recipients face risks of allogeneic responses against the graft, notably promoted by clinical infection. Our study did not have the statistical power to dissect the links between pneumotypes and different types of rejection, limited by the number of samples per rejection category in our dataset. Therefore, we grouped 29 samples from 17 patients with either CLAD, acute cellular rejection grade ≥2, or the presence of donor-specific antibodies (mean fluorescence intensity >1000), as these all indicate a suboptimal control of host immune competence and thus an increased probability of allograft injury (Fig. 6e, See "Methods" for clinical definitions). The majority of these samples were associated with Pneumotype$_{Staphylococcus}$ (41.7%) and Pneumotype$_{MD}$ (26.2%), followed by Pneumotype$_{Pseudomonas}$ (15.4%) and Pneumotype$_{Balanced}$ (13.1%), suggesting that this latter microbiota profile is associated with a lower risk of clinical complications. This was further corroborated by the count of circulating B lymphocytes in peripheral blood, suggesting more active humoral immunity in the presence of Pneumotype$_{Staphylococcus}$, Pneumotype$_{Pseudomonas}$ and Pneumotype$_{MD}$, compared to Pneumotype$_{Balanced}$ (Tukeys test, $P = 2.7 \times 10^{-2}$, $P = 2.6 \times 10^{-1}$ and $P = 1.2 \times 10^{-1}$ respectively; Fig. 6f). In addition to bacterial composition, anelloviruses were also linked to CLAD through a lower load of Gammatorquevirus (Wilcoxon test, $P = 7.0 \times 10^{-3}$, Fig. 6g), while no significant association was observed with Alphatorquevirus or Betatorquevirus (Wilcoxon test, $P = 1.5 \times 10^{-1}$ and $P = 9.0 \times 10^{-2}$ respectively, Fig. 6g).

Finally, we used the measurement of 'Forced Expiratory Volume in 1 s' (FEV1) to search associations between lung ecology and pulmonary function testing. This assessment was made irrespective of the diagnosis of CLAD, which requires an

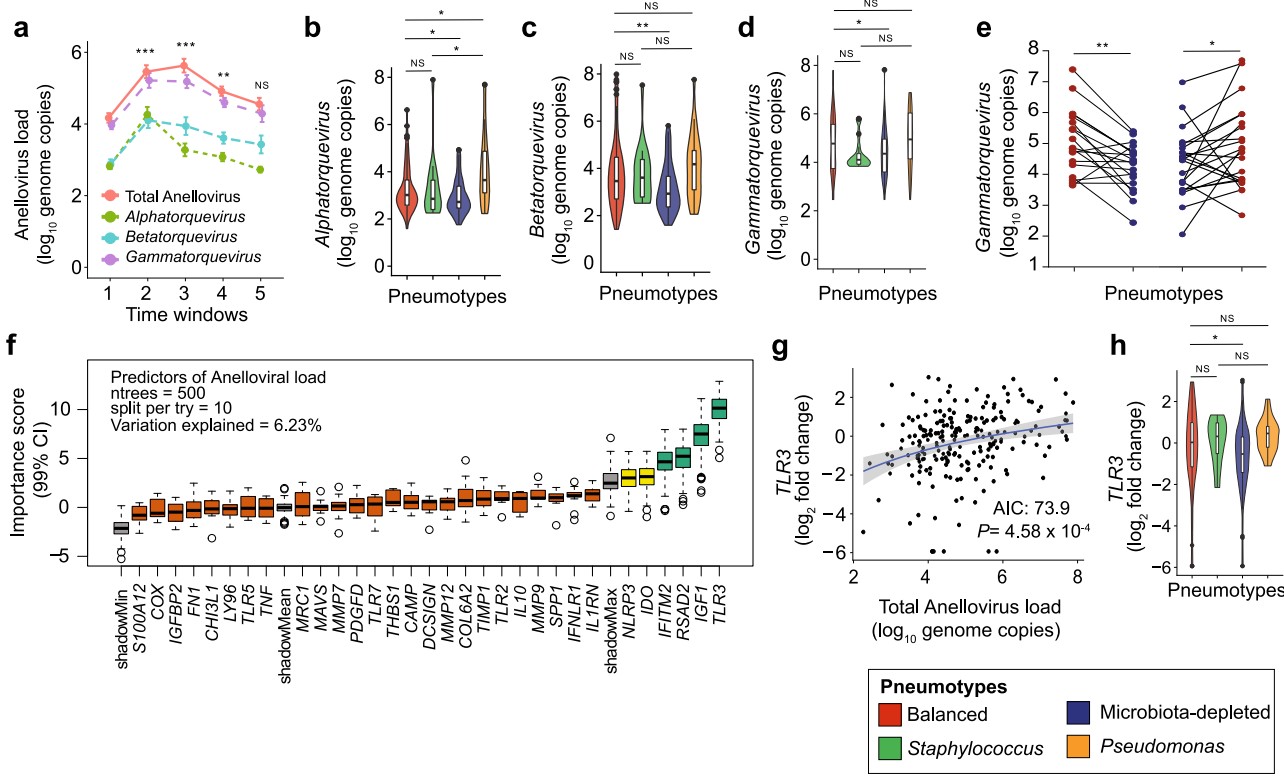

**Fig. 5 Anellovirus loads differ according to pneumotype and correlate with host physiology in the transplanted lung. a** Longitudinal progression of Anellovirus load (log$_{10}$ pan-Anelloviridae genome copies, pink) and its three major genera: *Alphatorquevirus* (green), *Betatorquevirus* (turquoise) and *Gammatorquevirus* (violet) over five time windows after transplantation. Data presented here as mean viral load (points) with error bars showing ±SD. Statistical significance is shown for total Anellovirus loads against time windows ($n = 225$, ANOVA, $F(5, 219) = 13.57$, two-sided, $P = 6.7 \times 10^{-10}$). **b–d** Violin plots show distribution of *Alphatorquevirus* ($n = 217$, Kruskal test, $\chi^2 = 17.04$, df = 3, two-sided, $P = 6.9 \times 10^{-4}$), *Betatorquevirus* ($n = 215$, ANOVA, $F(3, 211) = 4.57$, two-sided, $P = 3.97 \times 10^{-3}$) and *Gammatorquevirus* ($n = 216$, Kruskal test, $\chi^2 = 8.94$, df = 3, two-sided, $P = 3.0 \times 10^{-2}$) load across pneumotypes (plot colors). **e** Intra-individual analysis of *Gammatorquevirus* loads upon transition from Pneumotype$_{Balanced}$ (Red) to Pneumotype$_{MD}$ (Blue) (Wilcoxon test, two-sided, paired, $P = 1.7 \times 10^{-3}$) and vice-versa (Wilcoxon test, two-sided, paired, $P = 1.72 \times 10^{-2}$). Paired data (joined by black lines) presented here are viral genome copies (log$_{10}$, points). **f** Importance scores (99% Confidence Interval) of host genes in predicting anellovirus load analyzed by Random Forest regression model and Boruta feature selection. Predictor genes are categorized into Confirmed (Green), Tentative (Yellow) and Rejected (Orange), ntrees = number of decision trees, splits per try = number of random predictors sub-sampled and percent variance explained. **g** Scatter plots show correlation between expression (log$_2$ fold) of *TLR3* (stepwise regression AIC:73.9, ANOVA, $t = 3.56$, $P = 4.58 \times 10^{-4}$) with total anellovirus load (log$_{10}$ genome copies) across samples ($n = 231$). Linear regression is shown by the blue line with gray shaded area showing 95% confidence interval. **h** Violin plots show distribution of *TLR3* expression (log$_2$ fold) across the four pneumotypes ($n = 231$, Kruskal test, $\chi^2 = 10.66$, df = 3, two-sided, $P = 1.3 \times 10^{-2}$),. All box plots including insets show median (middle line), 25th, 75th percentile (box) and 5th and 95th percentile (whiskers) as well as outliers (single points). Post hoc analyses (95% Confidence Interval) were done by using Tukey's test (ANOVA) or Dunn's test (Kruskal test) with False Discovery Rate (FDR). * $P < 0.05$, ** $P < 0.01$, *** $P < 0.001$, NS = not significant.

irreversible drop in FEV1 below 80% of the baseline value, with prior exclusion of alternative confounding diagnosis (See "Methods"). Pneumotype$_{Staphylococcus}$ and Pneumotype$_{Pseudomonas}$ were associated with lower FEV1 values overall, with a frequent substantial decline below 80% predicted (Dunn's test, $P = 3.0 \times 10^{-2}$, Fig. 6h), while Pneumotype$_{Balanced}$, along with Pneumotype$_{MD}$, was linked to preserved lung function.

**Pneumotype$_{Balanced}$ shows the highest temporal stability and resilience in the transplanted lung.** Taking advantage of the longitudinal sampling, we explored the dynamics of the pneumotypes after transplantation. We analyzed transitions between pneumotypes in up to eight BALF samples per transplant, collected within five consecutive time windows (Fig. 7a). There was no significant difference in the distribution of pneumotypes across the different time windows ($\chi^2$ test, $P = 6.0 \times 10^{-1}$). Although most BALF samples were associated with Pneumotype$_{Balanced}$, transitions between two different microbiota pneumotypes occurred for about half of all consecutive sample pairs

(Supplementary Fig. 9). The transition dynamics were explained by Markov chain properties, i.e., the pneumotype of a given sample only depends on the state of the previous sample in the chain ($\chi^2$ test, $P = 3.3 \times 10^{-1}$, Fig. 7b). The transitions were irreducible, aperiodic and recurrent, and none of the pneumotypes behaved as an absorbing state (See "Methods"). Pneumotype$_{Balanced}$ exhibited the greatest stability, with the highest probability of recurrence (63%), fitting the Markov probabilities, followed by Pneumotype$_{MD}$ (42%; Fig. 7b). In addition, the large fraction of transitions toward Pneumotype$_{Balanced}$ between the first four time windows indicated a substantial resilience capacity for this profile. Accordingly, the transitions between Pneumotype$_{Staphylococcus}$, Pneumotype$_{Pseudomonas}$ and Pneumotype$_{Balanced}$ occurred mainly in the direction of this latter profile (Fig. 7b), while in contrast to model prediction, Pneumotype$_{Staphylococcus}$ and Pneumotype$_{Pseudomonas}$ appeared to be virtually disconnected.

Finally, we illustrate the relationship between the temporal dynamics of pneumotypes and clinical outcomes using a case study (Fig. 7c). Patient 35 diagnosed with pulmonary fibrosis

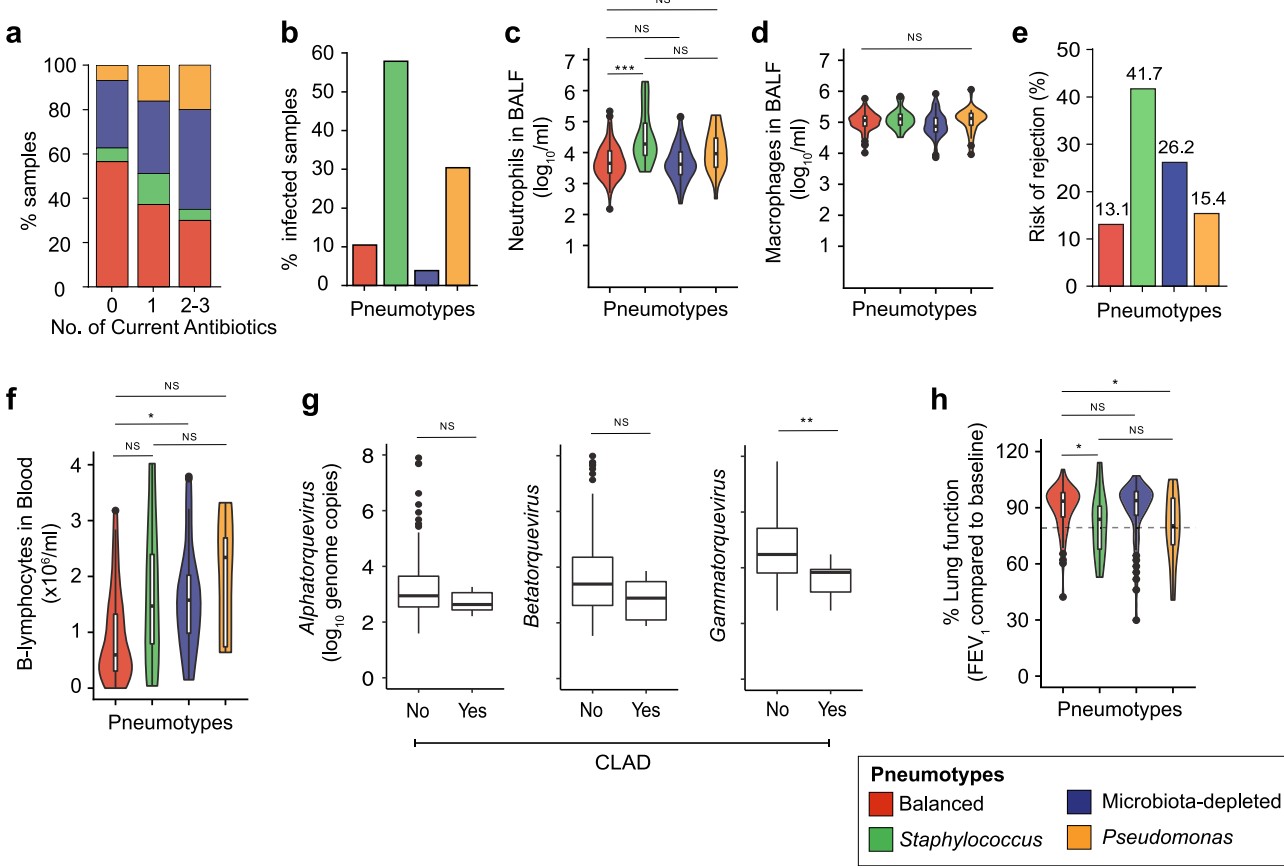

**Fig. 6 Association of post-transplant pneumotypes with pulmonary environment, local and peripheral host immunity and clinical status. a** Stacked bar plots showing proportion of samples associated ($n = 223$, Fisher's test, $P = 2.0 \times 10^{-3}$) with the four pneumotypes relative to the number of antibiotics administered. **b** Bar plots show the proportion of infected samples across four pneumotypes. Presence of infection was categorized as yes/no and statistical analysis was done by a generalized linear model, family = binomial, $n = 234$, Pneumotype$_{Staphylococcus}$; $P < 0.001$ and Pneumotype$_{Pseudomonas}$ $P = 1.6 \times 10^{-2}$, respectively; Fig. 6b) **c, d** Violin plots show distribution of Neutrophils ($n = 213$, ANOVA, $F(3, 209) = 11.72$, two-sided, $P = 3.95 \times 10^{-7}$) and Macrophages ($n = 224$, ANOVA, $F(3, 211) = 2.35$, two-sided, $P = 7.57 \times 10^{-2}$) counts ($\log_{10}$ cells per ml BALF) in lung linked to pneumotypes (plot colors). **e** Risk of rejection associated with each pneumotype (bar colors) was assessed by the cumulative percentages (%) of samples associated with each of the following conditions (See Methods): Chronic Lung Allograft Dysfunction (CLAD), presence of Donor-specific antibodies (DSA, Mean Fluorescence Intensity > 1000) or Acute cellular rejection (Biopsy score A2). **f** Violin plots show distribution of B-lymphocytes counts ($n = 87$, ANOVA, $F(3, 83) = 3.84$, two-sided, $P = 1.26 \times 10^{-2}$) in the blood associated with the four pneumotypes (plot colors). **g** Burden of three major anellovirus genera: *Alphatorquevirus* (Wilcoxon test, two-sided, $P = 1.5 \times 10^{-1}$), *Betatorquevirus* (Wilcoxon test, two-sided, unpaired, $P = 9.0 \times 10^{-2}$) and *Gammatorquevirus* (Wilcoxon test, two-sided, unpaired, $P = 7.0 \times 10^{-3}$) ($\log_{10}$ genome copies) in samples associated with CLAD (No or Yes, $n = 29$). **h** Violin plot showing distribution of lung function (% compared to baseline) measured by Forced Expiratory Volume in 1 s (FEV$_1$, $n = 206$, Kruskal test, $\chi^2 = 11.05$, df = 3, two-sided, $P = 1.1 \times 10^{-2}$) across four pneumotypes (plot colors). All box plots including insets show median (middle line), 25th, 75th percentile (box) and 5th and 95th percentile (whiskers) as well as outliers (single points). Post hoc analyses (95% Confidence Interval) were done by using Tukey's test (ANOVA) or Dunn's test (Kruskal test) with False Discovery Rate (FDR). * $P < 0.05$, ** $P < 0.01$, *** $P < 0.001$, NS = not significant.

received two transplants, providing 12 serial samples and presenting each of the four pneumotypes (Fig. 7c). Disruption of the Pneumotype$_{Balanced}$ occurred from month 25, followed by transition to Pneumotype$_{Staphylococcus}$ at month 30. This was accompanied by a positive culture for *Corynebacterium* spp. in line with our enrichment analysis and clinical culture tests (Supplementary Fig. 4, Fig. 3j), increased BAL neutrophilia, and a concurrent increase in host immune gene expression (heatmap in Fig. 7c). Thereafter, the patient was repeatedly exposed to antibiotics, and respiratory function started declining irreversibly leading to the diagnosis of CLAD at month 49 with Pneumotype$_{MD}$. Overall decrease in bacterial and anellovirus loads in the lung, suggested an increasing selection pressure on microbes most likely due to a combination of antibiotic treatment and poorly controlled host immune competence. The second transplant at month 50 was linked to a re-establishment of Pneumotype$_{Balanced}$, which aligned with preserved lung function, intermediate loads of

lung bacteria and anelloviruses, decrease in neutrophil counts and change in host immune gene expression. However, a transition to Pneumotype$_{Pseudomonas}$ was observed later until the end of sampling, with increased bacterial counts but no decrease in lung function (barplot in Fig. 7c). Taken together, these observations highlight the potential of integrating pneumotype with clinical and molecular data, with the primary goal of tracking disruption of Pneumotype$_{Balanced}$ beneficial to clinical stability.

## Discussion

In the current study, we capitalized on the availability of 234 longitudinal BALF samples from 64 lung transplant patients. We combined culture-dependent and -independent approaches to characterize the composition of the human lung microbiota, to obtain representative cultured isolates and test their growth

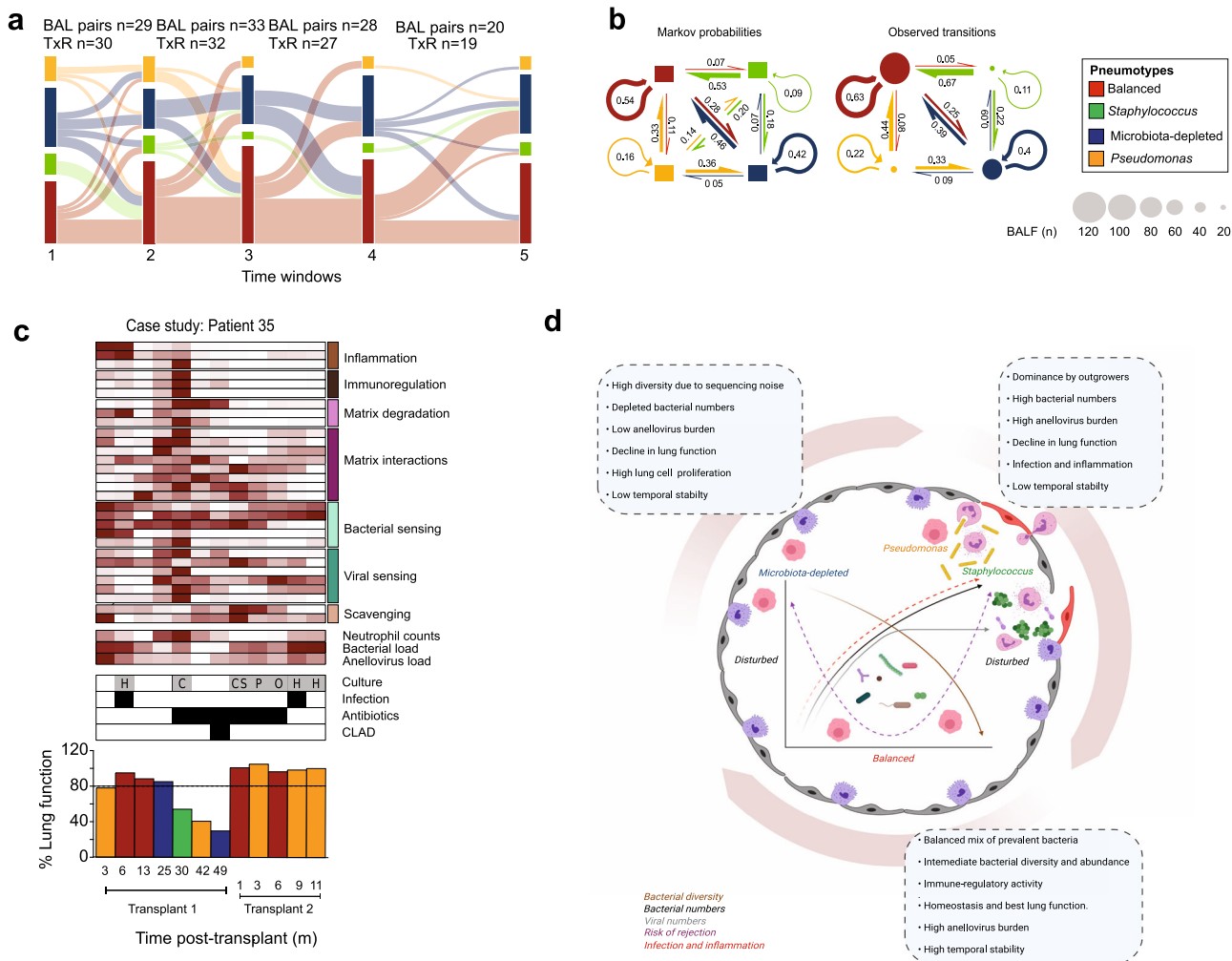

**Fig. 7 Longitudinal analysis of lung microbiota post-transplant and dynamics of pneumotype transitions. a** Sankey diagram showing transition of paired samples between pneumotypes (colors) across five time windows. **b** Markov chain model (See Methods) fitted to the observed pneumotype transitions (n = size of circle). Model was initiated with equal probabilities for each pneumotype (0.25, 100 bootstraps, left panel) and given transition matrix. Pneumotypes are represented by colored arrows/boxes, and the direction of a transition is indicated by a colored arrow of a thickness denoting the probability. **c** A patient case study showing transition of pneumotypes with clinical characteristics across two transplantation events. The heatmap shows host gene expression with functional categories (see also Fig. 4a, right vertical colored bars), neutrophil counts, bacterial and anellovirus loads in BALF across time and pneumotypes. Taxa obtained in routine clinical culture were abbreviated with letters. Samples positive for infection, ongoing antibiotic treatment or CLAD (black boxes) are presented above bar plots showing % lung function (see also Fig. 6g), across transplantation events and time post-transplantation (months) and pneumotypes (bar colour). **d** Scheme of bimodal disruption in lung ecosystem (colored arrows in a x–y plot) leading either to (i) a microbiota-depleted pneumotype with ambigous bacterial diversity (brown), low counts of bacteria (black) and viruses (gray), high lung cellular proliferation and chronic decline in lung function leading to rejection (purple), or (ii) pneumotypes dominated by opportunistic pathogens (*Staphylococcus* and *Pseudomonas*) with loss in bacterial diversity, high infection rate and inflammation (red), acute decline in lung function and rejection. Best-case scenario is defined by a middle ground with a balanced pneumotype consisting of the most prevalent bacteria in a homogenous composition with intermediate bacterial diversity, bacterial and viral abundance, high immune-modulatory activity and best preserved lung function. Original graphical art "Created using BioRender.com".

requirements, to assess the temporal dynamics of these communities in the lung, and ultimately to establish links with the host health status. In summary, our results show that the lung microbiota post-transplant is highly dynamic with a few predominant community members, many of which can be cultured, under different physiological conditions. We find that the lung microbiota post-transplant can be categorized into four compositional states, 'pneumotypes', based on distinctive bacterial community features. These pneumotypes have different temporal dynamics and bridge the gap between lung bacteria, anellovirus loads, host gene expression, and the physiological and immunological state of transplant recipients. Altogether, these results provide important advances in our understanding of lung

bacterial communities, their clinical significance, and the experimental tractability of major lung bacteria.

Our analyses show that the human lung microbiota post-transplant predominantly consists of oropharyngeal taxa similar to the microbiota of healthy lungs[5,6,26]. Hence, the presented results are not only relevant in the context of lung transplantation, but also provide general insights into the microbial ecology of the lower respiratory tract. Besides the high variability in taxonomic composition, we find that the total bacterial biomass in the lung can considerably vary between samples. Such quantitative differences in lung microbiota composition have also been found in previous studies. We find that a relatively small number of OTUs accounted for a large part of the total bacterial biomass

detected across all samples (19 OTUs contributing >75% of the biomass), despite the detection of more than 7000 OTUs. These included not only prevalent oropharyngeal taxa but also potential pathogens that outgrew in a few samples. These findings, and the fact that a key characteristic of Pneumotype$_{MD}$ is its association with low bacterial biomass, highlight the importance of considering absolute bacterial counts instead of relying only on proportional data in microbiome studies[40,48]. This is further evidenced by the fact that bacterial biomass can be predicted by host gene expression.

In addition to considering total bacterial biomass, demonstrating bacterial viability provides further evidence for the biological relevance of bacteria detected by sequencing. Previous studies have shown that bacteria from the human lung can be cultured with the majority growing under oxic conditions[6,13,14]. Our large-scale bacterial culturing approach, which included a wide array of culturing conditions, substantially expands the availability of bacterial isolates from the human lung and offers new insights about their phylogenetic diversity, physiological preferences, and metabolic potential. For instance, we show that many isolates, including prevalent community members, preferred to grow under anaerobic or in presence of 5% $CO_2$ conditions, suggesting the presence of regions with low oxygen concentration in the deep lung. Also, the culturing allowed us to phenotypically describe specific isolates in more detail and identify closely related pathogenic and non-pathogenic species of *Streptococcus*, which otherwise could not have been discriminated based on amplicon sequencing alone. Notably, the genus *Streptococcus* had the highest genetic, metabolic, and phenotypic diversity among all isolates, which may explain its presence throughout the human respiratory tract including sites with very different physicochemical properties[3]. We acknowledge that the presented culture collection is not exhaustive. We believe that this is due to the high variability of the lung microbiota and the fact that we have cultured a relatively small number of BALF samples, rather than the inability of some community members to grow in vitro under the tested conditions, or their non-viability in the lungs.

The observed differences in community composition and bacterial load between BALF samples suggest that the human lung microbiota is highly variable. However, our unsupervised machine learning approach identified four distinctive compositional states, Pneumotype$_{Balanced}$, Pneumotype$_{MD}$, Pneumotype$_{Staphylococcus}$ and Pneumotype$_{Pseudomonas}$. In a previous study on the lung microbiota of healthy individuals, a similar approach was used which resulted in the identification of two pneumotypes[5]. Strikingly, one of these previously identified pneumotypes was enriched in supraglottic taxa, i.e., mainly *Prevotella*, *Streptococcus*, and *Veillonella* resembling the Pneumotype$_{Balanced}$ from our study. The other pneumotype described by Segal et al. had similar characteristics as Pneumotype$_{MD}$ i.e., very low bacterial counts and a highly variable taxonomic composition. As with Pneumotype$_{MD}$ in our study, many of the taxa in this other pneumotype were considered to represent contaminations (or so-called background taxa). In contrast, Pneumotype$_{Staphylococcus}$ and Pneumotype$_{Pseudomonas}$ were not detected in this previous study, probably because it was based on a smaller cohort size and exclusively included samples from healthy individuals. *Staphylococcus*, the major community member of Pneumotype$_{Staphylococcus}$, is a frequent colonizer of the healthy human nasal mucosa capable of evolving on its host and switching to being infectious[49]. It has also been shown to dominate in neonatal lower airways, indicating potential early adaptation to human lung[4]. Together, these studies provide independent evidence for the existence of distinct compositional states of the human lung microbiota in different contexts. Moreover, the fact that the four pneumotypes are linked

to differences in host gene expression, bacterial and anellovirus loads, and allograft function and health state highlights their relevance, and suggests the existence of distinct ecological conditions in the lower respiratory tract, which are further discussed in the following sections.

We propose that Pneumotype$_{Balanced}$ is primarily associated with lung homeostasis (Fig. 7d), because it is characterized by a diverse bacterial community, with a moderate bacterial and viral load, and is linked to a human gene expression profile leaning toward immune modulation and peripheral immune tolerance. A particular characteristic of Pneumotype$_{Balanced}$ was the clear association with a high expression of Interferon-λ receptor 1 (*IFNLR1*), which suggests a link between the bacterial community and the maintenance of the epithelial barrier integrity[41] and antiviral defense[50]. Moreover, Pneumotype$_{Balanced}$ was linked to a down-regulation of Interleukin-1 receptor antagonist (*IL1RN*), produced in response to pro-inflammatory cytokines[44], and up-regulation of Interleukin-10 (*IL-10*), a tolerogenic cytokine[51]. This along with the previously reported association with Th17 immune response[9] indicates a possible role of Pneumotype$_{Balanced}$ in development of regulatory T cell and the maintenance of immune surveillance, as seen in case of gut bacteria[52,53]. In line with this, individuals with Pneumotype$_{Balanced}$ had the lowest risk of clinical complications at the time of sampling. Moreover, transitions from Pneumotype$_{Balanced}$ to other pneumotypes were the least frequently observed. Overall, these observations corroborate the steady-state associated with this pneumotype, suggesting that it is indicative of lung health and clinical stability after transplantation.

Pneumotype$_{MD}$ showed increased expression of the MRC1 gene, a characteristic of M2-like macrophages[42], and the Platelet-derived growth factor-D gene[45], an important contributor to airway remodeling. The low microbial load and the associated loss of a steady-state inflammatory level could be the underlying cause for the increased expression of these genes resulting in unrestrained host cell proliferation and increased deposition of extracellular matrix, as observed in CLAD[54]. Another striking feature of this pneumotype was the low expression of *TLR3*, a host gene involved in virus detection, which was consistent with the low loads of anelloviruses observed in the lung in the presence of this pneumotype. Virtually all lung transplant recipients carry anelloviruses, mainly in the plasma but also in the lung, with viral loads fluctuating over time[46,47]. Previous reports have shown that anellovirus counts in plasma are associated with host immuno-competence, infection and alloimmune rejection[47,55–59], suggesting that the Pneumotype$_{MD}$ is an indicator of a stronger selective pressure imposed by the host immune system on viruses and bacteria in the lung. This was confirmed by the low risk of infections and substantial risk of poorly controlled immune activity, highlighted by the high number of circulating B lymphocytes and either donor-specific antibodies, acute cellular rejection or CLAD.

A strongly contrasting pattern was observed for samples with either Pneumotype$_{Staphylococcus}$ or Pneumotype$_{Pseudomonas}$, which were both tightly linked to an inflammatory background. Here, viral and bacterial loads were increased relative to samples with Pneumotype$_{Balanced}$ and Pneumotype$_{MD}$. This was accompanied by a higher risk of infection and a consistent recruitment of neutrophils into the lung, ultimately leading to impaired pulmonary function (Fig. 7d). Notably, Pneumotype$_{Pseudomonas}$ was associated with low expression of Lymphocyte antigen (*LY) 96* / Myeloid Differentiation protein (MD2), an essential component of the human TLR4 complex[43]. Although it is tempting to associate the importance of *P. aeruginosa* in this pneumotype with a lack of engagement of the TLR4 pathway in the host[60,61], we cannot conclude about a causal link. In line with evidence that

infection activates alloimmune responses[62], samples with either Pneumotype$_{Staphylococcus}$ or Pneumotype$_{Pseudomonas}$ were also associated with a significant risk of poorly controlled immune activity and rejection.

Our study lacked sufficient statistical power required to explore the links between pneumotypes and different types of rejection (acute cellular rejection, antibody-mediated rejection, CLAD). However, grouping these samples allowed us to associate the Pneumotype$_{Balanced}$ with the lowest risk of poorly controlled immune activity. Furthermore, we could not assign causality to the observed links between the different constituents of the lung ecosystem. This was due to both the non-interventional nature of our approach and the multiplicity of confounding factors and their variability across the cohort. In particular, the underlying therapeutic treatments were expected to significantly modulate lung ecology, in addition to the effects due to infection and alloimmune response. This was illustrated by the observed link between Pneumotype$_{Balanced}$ and samples collected in the absence of ongoing antibiotic treatment, as opposed to the association between pneumotypes with disrupted bacterial communities and ongoing antibiotic therapy. Finally, follow-up studies are required to extend the knowledge gained by our single-site BAL sampling, which would not capture potential variability in ecological conditions across different regions of the lung[63] or between the lung and the upper respiratory tract[7,64].

In conclusion, our work provides a foundation for understanding the need for a balanced lung ecosystem along the bacterial community-viruses-host physiology axis, to maintain respiratory function and health. Overall, we propose that the four pneumotypes seem to follow the "Anna Karenina principle", where healthy communities vary little around a stable state, while perturbed communities are much more variable with unstable states[65]. We propose that the integration of multi-omics data analyzed using ecological principles will assist in the management and follow-up of lung transplant recipients, particularly with respect to CLAD prediction and supportive interventions. An important next step will be to establish causal links between lung ecology and allograft health by identifying the microbiota and host-related factors underlying pneumotype transitions. To this end, our bacteria collection LuMiCol provides a highly valuable resource that will serve as a foundation for future experimental studies using animal or cell culture models.

## Methods

### Study population, sampling and ethics statement

*Study design.* In this prospective longitudinal study, we used a cohort of 64 consecutive lung transplant recipients from the Organ Transplantation Centre of the University Hospital of Lausanne. We collected 234 BALF samples ($n = 1$–12 per recipient, mean 3.7) between 2 weeks and 49 months post-transplantation, during routine surveillance or clinically indicated bronchoscopies, from October 2012 to May 2018. Details on BALF collection and processing are provided below. We also included 11 negative control samples.

*Ethics statement.* The study was approved by the local ethics committee ("Commission cantonale (VD) d'éthique de la recherche sur l'être humain – CER-VD", protocol number 2018-01818), and all subjects gave written informed consent. Samples were anonymized according to local ethics committee requirements.

*Patient sample collection.* Patients underwent transoral bronchoscopy. For BALF collection, the bronchoscope was wedged either in the middle lobe or lingula of the allograft and 100–150 ml of normal saline were instilled in 50 mL aliquots that were pooled. BALF recovery was measured and the sample was submitted to cell differential determination according to routine clinical procedures. Two fractions of 3 ml were stored at 4 °C and centrifuged within 3 h at either 2000 or $14,000 \times g$ for 10 min, for future isolation of BALF cellular RNA and total DNA, respectively. Pellets were snap frozen, either directly or after cell lysis in RLT buffer (Qiagen, Hilden, Germany) to preserve RNA integrity, and were stored at −80 °C until further processing. A negative control obtained upon washing a ready-to-use endoscope with sterile saline was prepared following the same procedure.

### Bacterial culturing and establishment of lung microbiota culture collection (LuMiCol).

The lung microbiota culture collection (LuMiCol) can be searched on the GitHub page https://github.com/sudu87/Microbial-ecology-of-the-transplanted-human-lung by following the instructions in README.md. Bacterial strains can be requested via email to S.D. (sudip.das@unil.ch).

*BALF cultivation and archiving.* A volume of 100 μl of BALF was spread per plate of 15 different media (Supplementary Table 3) within 2–3 h following endoscopy. The plates were then immediately (within 2 h of extraction) incubated at the desired combination of oxygen and temperature conditions; aerobic (AE), 5% $CO_2$ ($O_2$: 17%, $CO_2$: 5%, Relative Humidity: 85%) and anaerobic (AN; $H_2$: 8%, $N_2$: 72%, O2: 40 ppm, $CO_2$: 20%) at a temperature range between 35 and 37 °C (Supplementary Table 3). Plates were incubated between 1 and 5 days. Bacteria were collected from plates by adding RPMI 1640 liquid medium supplemented with 15% glycerol and scraping using a Drigalski spatula and finally transferred into 96-well plates. Plates were made in triplicates for back up stocks. Each isolate was given a plate identifier (plate number - Px and well number - A1-H12) and a unique isolate code made with a combination of sample number, oxygen condition (AN/5% $CO_2$ /AE), Media used and isolate number (Supplementary Table 3, 5).

*Genotyping of bacteria.* Genotyping and species determination were based on PCR amplification of either universal 16S rRNA gene (V1-V5 region)[66] or specific marker genes, respectively. *Staphylococcus aureus* was identified by the amplification of *nuc* gene encoding the staphylococcal thermonuclease (Supplementary Table 7). The 16S rRNA gene sequences were aligned using two well curated databases containing high-quality 16S rRNA sequences to resolve species: SILVA SSU rRNA database and wherever SILVA failed to provide species identification we used the extended Human Oral Microbiome Database; eHOMD, http://www.homd.org. Phylogeny was performed by FastTree v 2.1.10 and visualized using iTOL.

*Bacterial growth determination by optical density.* Undefined rich media was represented by Todd-Hewitt (CM0189, Oxoid, UK) supplemented with yeast extract (0.5 g/L, LP0021, Oxoid, UK). RPMI 1640 medium with (11875085, ThermoScientific, USA) and without Glucose (11879020, ThermoScientific, USA) represented low complexity defined media. RPMI1640 without glucose was chosen as a proxy for deep lung fluids since it contains free amino acids, physiological salts, glutathione and no glucose, which are properties similar to lung epithelial lining[34].

One representative of each 47 phylotypes was revived on its individual isolation media (Supplementary Table 3), and bacterial biomass was scraped off the plates using 1X PBS. Bacterial suspension was diluted into 200 μL of appropriate media in 96-well flat-bottom plates (CytoOne®, CC7672-7596, Starlab, Germany). The plates were then immediately incubated at the desired combinations of oxygen and temperature conditions: aerobic (AE, 37 °C), 5% $CO_2$ ($O_2$: 17%, $CO_2$: 5%, relative humidity: 85%, 37 °C) and anaerobic (AN; $H_2$: 8%, $N_2$: 72%, O2: 40 ppm, $CO_2$: 20%, 34 °C) (Supplementary Table 3). Optical density was measured at 600 nm using a BioTeK Synergy H1 Hybrid Multi-Mode Reader starting from time day 0 (0 min) and every day (24 h) up to day 3 (72 h). Growth at each time point was calculated by the change in optical density from day 0 (ΔOD). The experiment was repeated three times and the median ΔOD for each day was used to create a heatmap.

### Species identification by phenotypic assays.

To differentiate staphylococcal and streptococcal species, primarily *S. aureus* from other staphylococci, and Viridans *Streptococcus* from pneumococcus, bacteria were screened for multiple phenotypes. As controls, *Staphylococcous aureus* ATCC 25904, *Streptococcus pneumoniae* strain D39; NCTC 7466 (pneumococcus control) and *Streptococcus mitis* NTCC10712 (viridans *Streptococcus* control, provided kindly by the group of Dr. Jan-Willem Veening, Lausanne, Switzerland) were used. For general overnight culture, streptococci were grown in Todd-Hewitt (CM0189, Oxoid, UK) supplemented with yeast extract (0.5 g/L, LP0021, Oxoid, UK) at 37 °C with 5% $CO_2$, 85% relative humidity and staphylococci were grown in Tryptic Soy Agar (CM0131b, Oxoid, UK) at 37 °C.

*Hemolysis detection on semi-solid agar.* For detection of hemolysis, bacteria were grown on Columbia agar (CM0331b, Oxoid, UK) supplemented with 5% defibrinated sheep blood (SR0051E, Oxoid, UK) and incubated at 37 °C under aerobic conditions or with 5% $CO_2$, 85% relative humidity and lysis of blood was observed after 24 h, after which complete hemolysis (beta-hemolysis) can be observed. The plates were then transferred to 4 °C for observing partial hemolysis alpha-hemolysis.

*High salt growth and mannitol fermentation test for staphylococci.* The ability of staphylococci to grow on high salt and ferment mannitol was tested by cultivation on Mannitol Salt Agar (MSA, 7.5% Sodium Chloride and D-Mannitol, CM0085B, Oxoid, UK) and incubation at 37 °C under aerobic conditions. This resulted in few combinations: Growth or no growth in MSA, growth in MSA but no fermentation of mannitol, growth in MSA and also fermentation of mannitol (designated by the conversion pink phenol red to yellow color).

*DNase activity assay.* Staphylococcal thermonuclease activity was tested by growing staphylococci on DNase agar (CM032, Oxoid, UK)[67]. Bacteria were grown overnight in Tryptic soy agar (16 h) and a single colony was picked and streaked across a straight line on DNase agar plate using a disposable plastic inoculation. Plates were incubated at 37 °C under aerobic conditions for 24 h, before flooding the plate with with 1 N HCl. After a dwell time of 30 s, the acid was drained out and a halo around the bacterial biomass indicated a positive result for DNase activity.

*Optochin resistance test.* For differentiating between viridans streptococci from *Streptococcus pneumoniae*, an optochin resistance test was performed[33]. Streptococci were grown on Columbia base agar (CM0331b, Oxoid, UK) supplemented with 5% defibrinated sheep blood for 24 h at 37 °C in presence of 5% $CO_2$ (SR0051E, Oxoid, UK). Colonies were then spread thoroughly on fresh blood agar plates with a cotton swab before placing optochin disks (74042, Sigma-Aldrich, Germany) on the center of the plates and incubated at 37 °C in presence of 5% $CO_2$ for 24 h. Inhibition zones were observed the next day for *S. pneumoniae* but not in case of *S. mitis*.

## BALF microbiota community analysis

*Bacterial 16S rRNA amplicon sequencing.* The total 16S rRNA gene copy number in BALF DNA was characterized by quantitative PCR using universal bacterial primers (Supplementary Table 7, which includes references). The bacterial community composition in BALF DNA was assessed using Illumina MiSeq sequencing with barcoded primers targeting the V1-V2 region (Supplementary Table 7). Amplification was performed using the Accuprime Taq DNA Polymerase High Fidelity kit (Invitrogen). Duplicate 20 µl PCR reactions consisted of 2 µl of 10× Accuprime buffer II, 0.44 µl of each F-27 and barcoded R-338 primers at 10 mM, 9.03 µl of ultrapure water, 0.09 µl of AccuPrime Taq DNA Polymerase and 8 µl of DNA template with the following cycling parameters: initial denaturation 3 min at 94 °C, followed by 40 cycles of 30 s denaturation at 94 °C, 30 s annealing at 56 °C, and 90 s elongation at 72 °C, with a final extension at 72 °C for 5 min. Amplicons were quantified using a LabChip GX instrument with DNA 1 K kit (Perkin Elmer), pooled at equimolar amounts and purified using AMPure XP bead cleanup system (Beckman Coulter). Libraries were then diluted to 12 pM and spiked with 25% phiX before loading on the Illumina MiSeq platform using paired-end chemistry, generating $250 \times 2$ read lengths.

*Data processing and OTU picking.* Data curation and analysis was performed using a custom pipeline[32] (see "Data and code availability"). The major packages used are described in the "Software used and statistical analysis" section. Primers were removed and reads were joined with fastq-join with a minimum overlap of ten base pairs, demultiplexed and quality filtered (PHRED score $Q > 28$ in 75% of read length). Sequence quality was assessed using FastQC and the first 75 bases were trimmed using fastx_trimmer. Both raw (All_BAL_samples_raw_fastqc) and processed (All_BAL_samples_processed_fastqc) sequence quality analyses are available along with other datasets https://doi.org/10.5281/zenodo.4556025.

Singletons were removed using vsearch. Prior to OTU picking, taxa were clustered into centroids with >98% identity, chimeras removed, and the data were mapped to the centroids with >97% coerced into a single OTU. The sequences obtained after OTU picking were further used for alignment and taxonomy using the SINA aligner and the SILVA database release SSURef_NR99_132_SILVA_13_12_17. Phylogeny was performed by FastTree v 2.1.10 and visualized using iTOL.

Before proceeding further, we removed OTUs with ambiguous taxa i.e., Phylum = "NA" and any samples that had <$10^4$ reads and no information on bacterial abundance. In conventional practice, samples with low biomass (detected by 16S qPCR) are excluded but we elected to retain them if sequencing was successful ($\geq 10^4$ reads). However, in these low biomass samples the risk of obtaining spurious taxa is higher, requiring comparisons with negative controls, which included Bronchoscope pre-wash, DNA extraction reagents and no-template PCR reaction. We found that negative control samples contained 1015 OTUs, including those from family *Enterobacteriaceae* and genera *Limnohabitans, Fodinicola, Staphylococcus, Flavobacterium, Cutibacterium, Acidovorax, Tepidimonas* and *Variovorax*. After quality filtering and rarefying the data to the sample with the lowest number of reads, we ended up with 7164 OTUs at 97% identity in 16S rRNA gene. These OTUs belong to 37 phyla with the most abundant phyla being *Bacteroidetes, Firmicutes, Proteobacteria* and *Actinobacteria*.

*Extraction of OTU-isolate match pairs.* In order to assign the cultured bacteria to OTUs (i.e., OTU-isolate matching pairs), we aligned the high quality 16S rRNA gene sequences obtained by Sanger sequencing from 215 LuMiCol isolates (phred score: Q30 > 90%) to the dereplicated 16S rRNA amplicons obtained from the Illumina sequencing using MAFFT alignment tool in Geneious Prime software (Geneious, New Zealand). The LuMiCol sequences were then trimmed to same size as the 16S rRNA amplicons and included in the community analysis as new sample "LuMiCol" after the dereplication step[32] (–usearch global; Step 13- Das_et_al_2020_analysis_pipeline_1.md; https://github.com/sudu87/Microbial-ecology-of-the-transplanted-human-lung) for OTU and taxonomic assignment (See also "Data processing and OTU picking"). OTUs containing Sanger sequence reads from the sample "LumiCol" were identified and flagged as OTU-isolate matching pairs.

*Prevalence and absolute abundance analysis.* Prevalence was informed by the incidence of each OTU across all samples in the cohort. This was calculated by using the function amp_core from ampvis2 v 2.3.2. The output table (Supplementary_Data_3) consists of serial number, OTU number, frequency (overall), frequency at 1% relative abundance (freq_A), abundance (mean relative), and taxonomy. Absolute abundance was calculated by using the phyloseq object with the relative abundance OTU table and multiplying each OTU in each sample by the 16S rRNA gene copies detected per ml of BALF sample (see section "Bacterial 16S rRNA amplicon sequencing").

*Alpha diversity analysis.* Alpha diversity indices were obtained from Rényi diversity and corresponding Hill numbers using the function 'renyi' from the *vegan* package in R. The Hill numbers calculated were $H_0$ (Number of species), $H_1$ (exponent of Shannon diversity), $H_2$ (Inverse Simpson) and $Hill_\infty$ (Berger-Parker index i.e., $1/\max\ p_i$ (inverse of diversity of order infinity). Proportion of dominant OTUs ($\max\ p_i$) was calculated by $1/Hill_\infty$ (maximum proportion of species $i$).

*Beta diversity analysis.* Beta diversity was calculated by applying the 'distance' function on the phyloseq object. Presence/absence of OTUs was calculated by using Sørenson's index, which was interchangeably used with the term Bray–Curtis distance (binary = TRUE, calls for function 'vegdist' from vegan package). For species abundances, Morista-Horn distance measure (calls for function 'vegdist' from vegan package and uses the distance measure 'horn') was used. Statistical analysis of beta diversity was peformed by PERMANOVA with the *adonis* function in the vegan package in R and multiple comparison was performed by using the wrapper function pairwise.adonis. We used 10,000 permutations as standard for all our comparisons.

*Enrichment analysis of OTUs.* Similar to prevalence analysis across the entire cohort, enrichment analysis of OTUs was repeated for individual PAMs. The amp_vis object was split into PAM groups, and the incidence percentages were then calculated by using the function amp_core in ampvis2 package. Prevalence of each OTU in individual PAMs was compared to the entire cohort and the 30 most prevalent and/or abundant microbiota members were plotted as described in Supplementary Table 2.

For enrichment analysis of OTU abundances, each PAM was compared to a file containing absolute abundances of the other 3 PAMs. Statistical analysis was performed by ART-ANOVA, with a two factorial design (group = single PAM vs other 3 PAMs, variable = OTU IDs). Marginal means were calculated by using the emmeans R package. Pairwise differences were calculated followed by Benjamini–Hochberg multiple testing for False Discovery Rate (FDR). Plotting was limited to the 30 most prevalent and/or abundant microbiota members as described in Supplementary Table 2.

## Quantitative analysis of host gene expression and anellovirus load in BALF

*BALF cellular RNA extraction and real-time quantitative PCR for gene expression analysis.* BALF cell lysates were transferred into a QIAshredder column (Qiagen) for homogenization, and total RNA was extracted using RNeasy Mini Kit (Qiagen) according to the manufacturer's instructions. RNA concentration was determined using a Nanodrop ND-1000 spectrophotometer (Thermo Fisher Scientific, Waltham, MA, USA) and reverse transcription was performed using iScript cDNA Synthesis Kit (Bio-Rad, Hercules, CA, USA). Characterization of BAL fluid cell gene expression profiles was based on multiplex real-time PCR analysis using custom oligonucleotide primers and probes (Microsynth, Balgach, Switzerland) for a set of 31 genes (Supplementary Table 7). We used guanine nucleotide-binding protein, beta polypeptide 2-like 1 (GNB2L1) gene as a reference gene, given its high expression stability in BALF cells in both health and disease[68]. Amplification was carried out using iQ Multiplex Powermix Master Mix and a CFX96 Real-Time detection system.

For radar chart visualization, the samples were sorted according to their association with one of the four pneumotypes, and the median expression values for each gene were determined within each group. For each gene, the highest median was then arbitrarily set to 1 and plotted as the maximum value in the corresponding chart. The median values obtained within the other groups were normalized accordingly.

*Quantification of anellovirus load.* Based on the tropism of *Anelloviridae* for hematopoietic cells, we quantified the load of this virus family starting from the DNA extracted from the BALF cellular pellet. Absolute quantification of pan-Anelloviridae, *Alpha-, Beta-* and *Gammatorquevirus* (Supplementary Table 7) was performed using the CFX96 Real-Time detection system (Bio-Rad) based upon values obtained with a set of purified amplicons used as standards.

## Machine learning and statistical modelling

*Unsupervised learning for Pneumotype discovery.* Pneumotypes were obtained by running $k$-medoid-based unsupervised machine learning on a Bray–Curtis dissimilarity matrix (binary = FALSE) using Genocrunch (see Section "Software used and statistical analysis"). The program utilizes the *pamk* function of the R package fpc version 2.1.10 to cluster samples while optimizing the number of clusters based

on the average silhouette width[69,70]. Input and output parameters used in Genocrunch were provided in the corresponding json files, which are available on the GitHub repository (https://github.com/sudu87/Microbial-ecology-of-the-transplanted-human-lung). In addition, detailed input and output files from Genocrunch are contained in Supplemetary Data.

*Optimization of random forest parameters*. We used Random Forest to predict (i) the pneumotype and (ii) the bacterial and viral counts based on host gene expression data. The median normalized expression levels of the 31 analyzed host genes presented in Fig. 4a from all 234 subject samples were used as predictors. For predicting pneumotypes, we used a classification model and for predicting bacterial or viral loads, we used a regression model.

While machine learning algorithms typically use a training and a prediction set, Random Forest allows for cross-validation, i.e., random shuffling of existing data, allowing predictions to be made without the need for an external prediction data set. Such cross-validation was first performed ten times and then iterated three times. We optimized the models using the 'caret' package in R, by tuning the two most important parameters: (i) the mtry (i.e., splits per try—number of randomly selected predictor genes to be sampled at each iteration), and (ii) the ntrees (i.e., number of decision trees), using a specific mtry at each iteration. Full details on the entire optimization process and the code can be found on the GitHub repository (https://github.com/sudu87/Microbial-ecology-of-the-transplanted-human-lung/blob/master/Das_et_al_2020_analysis_pipeline_2.md) from Steps 56–58. In short, to optimize mtry in case of pneumotype prediction, we used the 'grid search' function (tenfold cross-validation, with three iterations of the whole process). For regression models, we used the 'random search' function, whose 'tuneLength' argument randomly specifies the number of mtry values, which we set to the maximum of 30. We found that mtry of 5, 21 and 10 gave the best accuracy results for pneumotype, bacterial counts, and viral count prediction respectively. Using these values, we optimized the number of decision trees (ntrees) needed for good predictions, by running the 'grid search' or 'random search' functions within an incremental loop to vary the ntrees from 500 to 5000 for both the classification and regression models. For the classification model predicting pneumotypes, we found little difference in accuracy and sensitivity percentages. Hence, we kept the minimum number of trees i.e., 500.

For regression models predicting bacterial and viral counts, instead of accuracy or sensitivity percentages, the results were assessed by low Root Mean Squared Error (RMSE) and high regression value i.e., R2. Similarly, we observed little difference in RMSE and R2 with the series of ntrees tested, and used ntree = 1000.

After every analysis, random forest provides results in terms of error rate (out-of-box error) and table for the classification predictions (Supplementary Table 5) and percentage of variance explained for regression predictions (Figs. 4i and 5f). This indicates the importance of each feature, i.e., gene predictor, which is defined by the increase in accuracy (in classification models) and variances (in regression model) at every step in a decision tree, while using a particular feature amongst others. 'Boruta' algorithm creates several copies of the data where it randomly shuffles each feature. These permuted features (shadow features) are attached to existing data, on which the random forest function is then applied. At every iteration during a decision tree, the algorithm compares the original features to a threshold, which is created by recording the highest importance within the shadow features (i.e., shadowMax). Additional thresholds are also created, shadowMin, i.e., the minimum importance within the shadow features, and shadowMean, i.e., the mean importance within the shadow features. The features that pass this threshold i.e., having higher Z score than the shadowMax are assigned as "Confirmed" while other features are "Rejected". However, since a random forest algorithm runs for a limited number of steps, features remain that pass the threshold but cannot be confirmed and are hence assigned as "Tentative". This can be handled by using the function 'TentativeRoughFix' that fills the missing decisions by comparing the median Z score of a feature to the median of the most important shadow feature (i.e., shadowMax).

*Correlations of gene expression with predicted features by random forest regression*. Gene predictors from random forest analysis with importance scores higher than 10 were further fitted into additive linear models with either bacterial or viral copy numbers i.e., lm(copy number ~ gene A + gene B). The best models were selected with the stepAIC function of the MASS package in R, which performs a stepwise model selection by AIC (Akaike Information Criteria).

**Clinical measurements and definitions**. Determination of the cell differential in the BALF, B-cell count in peripheral blood by mass cytometry, and bacterial culture for diagnostic purposes were performed according to in-house routine clinical procedures.

*Definition of acute bacterial infection*. Acute bacterial infection was defined as positive BALF culture with dedicated antibiotic treatment, associated with clinical signs and symptoms, such as a decrease in FEV1, new or progressive infiltrate on standard chest radiography or CT-scan, fever, positive pulmonary auscultation, cough, dyspnea, hemoptysis, pleuritic pain, purulent sputum.

In contrast, a BALF culture positive for a pathogen, but not associated with the administration of antibiotic therapy and without clinical signs and/or symptoms, was considered as a bacterial colonization and not as an acute bacterial infection.

*Definition of chronic lung allograft dysfunction (CLAD)*. CLAD was defined as a loss of more than 20% of the expiratory volume in 1 s (FEV1) of the mean of the two best values (i.e., the baseline FEV1) since transplantation, without other obvious cause and without reversibility, in accordance with the diagnostic criteria specified by the Pulmonary Council of the International Society for Heart and Lung Transplantation[54]

**Software used and statistical analysis**. Various statistical approaches and tests were used depending on the analysis, as detailed in the appropriate sections. All analyses were performed in R version 3.5.2, python v 2.6 and bash on macOS Mojave v 10.14.6.

Citations were included for all software used except for packages available via CRAN Repository and tools that are available via downloads from public database. For sequencing quality control and curation, FastQC (https://www.bioinformatics.babraham.ac.uk/projects/fastqc/) and FASTX-Toolkit (http://hannonlab.cshl.edu/fastx_toolkit/index.html) were used. A custom pipeline for 16S rRNA gene amplicon sequence analysis was built using QIIME v1.9[71], vsearch v 2.3.4[72], ampvis2 v 2.3.2[73], phyloseq 1.26.1[74] and vegan package version 2.5–6. Alignment and taxonomic classification were obtained using SINA aligner (https://www.arb-silva.de/aligner/) on the local computer. Phylogeny was performed by FastTree v 2.1.10 and visualized using iTOL (https://itol.embl.de). *K*-medoid-based unsupervised machine learning was performed on Genocrunch (www.genocrunch.epfl.ch).

Differential abundance was tested using ART-ANOVA from ARTool package version 0.10.7. Machine learning classification and regressions were performed with the *randomForest* package and its wrapper algorithm Boruta for feature selection[75]. For optimization of random forest parameters, the *carat* package was used in R[76].

Data normality and homogeneity of variances were tested using Shapiro–Wilk test and Levene's test. LuMiCol isolate sequences were curated using Geneious Software v 10.2.6, New Zealand. All graphical illustrations are original art created using BioRender Science Suite (https://www.biorender.com) without using any templates and exported under paid academic subscription.

**Reporting summary**. Further information on research design is available in the Nature Research Reporting Summary linked to this article.

## Data availability

Raw sequencing data from all samples used in the study were submitted to Short Read Archive, National Center for Biotechnology Information under the BioProject PRJNA632552 and BioSample accession SAMN14911405.

Quality control reports and supplementary data have been made available on zenodo at https://doi.org/10.5281/zenodo.4556025. For searching and requesting bacterial isolates from the lung microbiota culture collection (LuMiCol), refer to the GitHub page https://github.com/sudu87/Microbial-ecology-of-the-transplanted-human-lung and follow the instructions. Bacterial strains can be requested via email to S.D. (sudip.das@unil.ch) and will be sent free of charge and shipping costs will be paid by the receiver. This is also subject to a Materials Transfer Agreement (MTA) set by PACTT (Powering Academia-industry Collaborations and Technology Transfer), a joint technology transfer office of the University of Lausanne (UNIL) and the University Hospital of Lausanne (CHUV). Website: https://www.pactt.ch.

## Code availability

All the custom scripts have been made available on zenodo at https://doi.org/10.5281/zenodo.4556025 and linked to GitHub.

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

## Acknowledgements

We thank the clinicians of the Department of Respiratory Medicine of Lausanne University Hospital (CHUV) who performed the bronchoscopies, the clinical database team of the Thoracic Surgery Department of CHUV (Prof. Thorsten Krueger, Audrey Roth RN and Fébronie Maillefer RN) for contributing in the completion of the clinical dataset, Julie Pernot for technical assistance, and the transplant patients for allowing their clinical data and BALF samples to be used for clinical research.

## Author contributions

Conceptualization, S.D., E.B., D.-A.W., J.-D.A., B.J.M., P.E. and L.P.N.; Data Curation, S.D., E.B., A.K., J.-D.A. and V.T.; Formal Analysis, S.D., E.B. and D.-A.W.; Funding Acquisition, S.D., A.K., C.V.G., B.J.M., P.E. and L.P.N.; Investigation, S.D., E.B., D.-A.W., V.T., J.-D.A., M.-F.D., L.M. and C.P.; Methodology, S.D., E.B., D.-A.W., V.T. and G.B.-R.; Project Administration, E.B., A.K., B.J.M., P.E. and L.P.N.; Resources, A.K., C.P., A.R., C.V.G., B.J.M., P.E. and L.P.N.; Software, S.D. and G.B.-R.; Supervision, E.B., A.K., G.B.-R., C.V.G., B.J.M., P.E. and L.P.N.; Validation, S.D., E.B., D.-A.W., V.T., J.-D.A. and P.E.; Visualization, S.D., E.B. and D.-A.W.; Writing - Original Draft, S.D., E.B., and P.E.; Writing - Review & Editing, S.D., E.B., A.K., D.-A.W., V.T., G.B.-R., J.-D.A., M.-F.D., L.M., C.P., A.R., C.V.G., B.J.M., P.E. and L.P.N.

## Funding

The project was funded by a Marie Sklodowska-Curie Individual Fellowship (awarded to SD, "HUMANITY", Grant no. 800301), an Interdisciplinary grant from the Faculty of Biology and Medicine of the University of Lausanne (awarded to P.E. and B.M., Grant no. 26075716), an ERC StG (awarded to P.E., "MicroBeeOme", Grant No. 714804), two Swiss National Science Foundation grants (awarded to P.E., Grant no. 31003A_160345 and 31003A_179487), a HFSP Young Investigator grant (awarded to P.E., Grant no. RGY0077/2016), a grant from the Swiss Lung Association (awarded to A.K., Grant no. 2018-16), a "Pépinière" grant from the Faculty of Biology and Medicine of the University of Lausanne (awarded to A.K.), and a GlaxoSmithKline educational grant (awarded to A. K.). E.B. was supported by "Fondation Professeur Placide Nicod". B.J.M. is a VESKI innovation fellow and an NHMRC Senior Research Fellow.

## Competing interests

The authors declare no competing interests.
