## [Peer Review File · Nature Communications]

REVIEWER COMMENTS

Reviewer #1 (Remarks to the Author):

This study by Das et. al will be a tour de force in the lung transplantation microbiome field. It is an extremely strong study that also includes longitudinal analyses. The manuscript highlights many features of host-microbiome interactions and uses high throughput amplicon sequencing, culture, and qPCR. In addition, this study also provides host response data and then ties all the data together using various bioinformatic approaches. It is a rare study that provides such a broad interdisciplinary approach in human studies. It is easy to read and the message is very clear. Overall, this is an excellent paper and a solid advance in the field of the lung microbiome, transplantation biology and human microbial ecology.

Reviewer #2 (Remarks to the Author):

The manuscript by Das et al investigates the lung microbiome using a combination of culture-independent bacterial profiling and bacterial culture on a subset of samples. This is couple with extensive host (and viral) analysis as well. The study includes 234 samples from 64 patients which is sufficiently large group to begin to stratify the microbiomes with patient characteristics. This is a strong study with mew insights into the post transplant lung environment.

While I am enthusiastic about this study there are a number of major and minor issues that need to be addressed.

Major Issues:

The microbiome analysis is using rather dated methods and rather than OTUs the current best practices used amlicon sequence variant approaches (dada2 or deblur) which also implement error correction and reduce sequence error noise in the data. This also has better resolution that OTU clustering. Not using ASV based approaches does require a strong justification.

in addition the comparison of the isolates 16S sequence to the culture independent data was also done by clustering, the use of ASVs would allow a more precise comparison. Clustering reduces resolution which is an advantage of having cultured isolates in the first place.

Line 297 - "semi quantitative culturing" Why is this not quantitative? I recognize the difficulty with BALF but all of the data is generated from this and it just as semi-quantitative. It would be of great value to compare the CFU recovered to the 16S qPCR for bacterial load. A fundamental difficulty in the lower airways in culture-independent methods is the low turnover of DNA that accumulates and this gets proportionally worse with disease.

line 537 - "537 The high variability in community composition and bacterial load between BALF samples may suggest that the human lung microbiota is highly erratic." This is a very unusual statement- and I am not sure what the authors are implying.

A major claim is that this is one of only a few studies to apply culturing to the lung microbiome. (line 83 and elsewhere e.g line 234). This ignores a large body of work from chronic airway disease where this has been done quite extensively (and more comprehensively than reported here). While less has been published on the healthy lung microbiota, I would point out that the transplanted lung is not equivalent to a healthy lung and caution should be taken in interpreting it as such.

It would have been informative to have more longitudinal analysis. Although this is done in an overview of the pneumotypes in Fig 7, this is a missed opportunity as there is regular sampling at the same time intervals in over half the samples (eg 3,6,12,24 months)

There was no discussion of pre-transplant patient infections. The upper respiratory tract is a reservoir for lower airway pathogens and most transplant patients in chronic airway disease (particularly CF and bronchiectasis) that get re-infected are by the same STRAIN they were previously infected by. Do the pneumotypes defined by Staphylococcus and Pseudomonas respectively represent patients that had these infections prior to transplant?

Minor issues:

line 108-109. The problem with these types of analyses/generalizations is the talking about these at the genera level. While there are large numbers of Streptococcus species that differ dramatically in their pathogenic potential (a Group B strep is hardly equivalent to a S salivarius).

line 239 It is well known that anaerobic growth of Pseudomonas requires nitrate (or another terminal electron acceptor) this is sufficient nitrate present in chronic airway disease to support anaerobic growth of Pseudomonas and there is data to suggest that it does grow under anaerobic conditions.

242 "...these results indicate towards changes physicochemical conditions in the lung that may favor the growth of aerobic

bacteria with potentially pathogenic properties." This is a rather broad statement that I do not think is very accurate. These are two opportunistic pathogens that are the most common to infected patients with chronic airway disease, and must be well adapted to do this not just any "aerobic bacteria with potentially pathogenic properties" which is a pretty long list.

The authors do seem to use strain and species interchangeably and should be cautious not to. e.g. lines 525-527. While most species can be resolved by 16S (not all but most even in the streptococci) just not always with short amplicons, it is exceptional to resolve strains by the 16S gene. There are varying degrees of pathogenicity between strains within a species - it is not clear if that is what they are referring to here or species resolution.

line 535 - numerous studies have demonstrated most of these organisms are readily cultured with very few exceptions in the airways. It could be the choice of culture conditions chosen here which may have limited their success.

line 551-552. Staphylococcus is one of the most common lower respiratory pathogens in chronic airway disease and not uncommon as an acute respiratory pathogen.

In the culture conditions table, 5%CO₂ is described as micro-aerobic. It is not.

As a final point, which will know doubt be contentious but highly relevant (and I will not categorize as a major or minor issue but just a comment). I do have a fundamental problem with the concept of describing the microbiota of chronic airway disease as a dysbiosis. Many in the field (and the best data) argues that the healthy lung is transiently colonized (with very little growth) with a very low biomass and cleared. This transient colonization is no doubt an important immune modulator of the lung. But this is very different that the chronic colonization with orders of magnitude higher bacterial burden that is actively growing in the lower airways. This is not really the dysbiosis as described at other body sites which have resident growing communities of microbiota (not that the term dysbiosis is fraught with problems of its own...).

Reviewer #3 (Remarks to the Author):

The following comments are focused (as requested by the editor) on the data analysis and the ML aspects.

The main problem of this work is rarefaction. Looking at the text and code it seems that no rarefaction was performed on the microbiome data.

While rarefying is not an optimal normalization method, as it reduces the statistical power, it is essential to reduce the artifacts due to the differences in library sizes across samples (doi:10.1186/s40168-017-0237-y). Authors should therefore apply rarefaction (whithout replacements) before alpha/beta, statistical and machine learning analyses in order to obtain meaningful results.

Other points

Data processing and OTU picking:

- Line 818: What did you mean by : "... as this increases the risk of obtaining spurious taxa, we removed OTUs with ambiguous taxa i.e. NA ..."? Did you remove any unclassified OTU? At which taxonomic level? Please explain this point better.

- Line 825: "... After quality filtering and normalization, ..." Which normalization?

Extraction of OTU-isolate match pairs:

- This part is difficult to understand. Please rewrite it in a professional and detailed form.

Unsupervised learning for Pneumotype discovery:

- Genocrunch parameters and detailed output are missing.

Random Forest:

- It is not clear how the models were optimized for best accuracy. Please write the section as detailed and clear as possible.

- Line 935: Please explain how importance scores are computed.

- Lines 939-941: "Gene predictors from random forest analysis with importance scores more than 10 were

further fitted into additive linear model with either bacterial or viral copy numbers ..."

Please explain why this makes sense since the RF classifier is strongly non linear.

Please explain why you have taken predictors with importance scores more than 10.

Code availability:

- The code should be uploaded to public code repositories, like Github or Bitbucket.

Point-by-point reply to reviewer comments:

Reviewer #1 (Remarks to the Author):

This study by Das et. al will be a tour de force in the lung transplantation microbiome field. It is an extremely strong study that also includes longitudinal analyses. The manuscript highlights many features of host-microbiome interactions and uses high throughput amplicon sequencing, culture, and qPCR. In addition, this study also provides host response data and then ties all the data together using various bioinformatic approaches. It is a rare study that provides such a broad interdisciplinary approach in human studies. It is easy to read and the message is very clear. Overall, this is an excellent paper and a solid advance in the field of the lung microbiome, transplantation biology and human microbial ecology.

Reply: We thank the reviewer for this very positive assessment of our work. We are happy to see that it is so well received and share the reviewer's excitement about the advancement for the field.

Reviewer #2 (Remarks to the Author):

The manuscript by Das et al investigates the lung microbiome using a combination of culture-independent bacterial profiling and bacterial culture on a subset of samples. This is couple with extensive host (and viral) analysis as well. The study includes 234 samples from 64 patients which is sufficiently large group to begin to stratify the microbiomes with patient characteristics. This is a strong study with mew insights into the post transplant lung environment.

While I am enthusiastic about this study there are a number of major and minor issues that need to be addressed.

Reply: We thank the reviewer for the positive feedback and the thorough review of our study.

Major Issues:

The microbiome analysis is using rather dated methods and rather than OTUs the current best practices used amlicon sequence variant approaches (dada2 or deblur) which also implement error correction and reduce sequence error noise in the data. This also has better resolution than OTU clustering. Not using ASV based approaches does require a strong justification. in addition the comparison of the isolates 16S sequence to the culture independent data was also done by clustering, the use of ASVs would allow a more precise comparison. Clustering reduces resolution which is an advantage of having cultured isolates in the first place.

Reply: Thank you for raising this important point. It is true that ASV-based approaches more accurately describe single sequence variants than OTUs. However, it is also true that this approach has been more recently developed and hence hasn't yet been as meticulously benchmarked as OTU-based approaches which exist for over 20 years. While we acknowledge that ASV-based approaches are currently more popular and seem to replace other methods, OTU-based approaches are neither wrong nor outdated. Which method to implement really depends on the specific questions and hypothesis to be tested.

For example, the phylogenetic level at which ecological and evolutionary processes affect organisms is variable between bacterial taxa and environments. Phylum-level ecological responses are observed in mouse gut communities in response to some diets, but strain-level responses are observed in phage predation or species-species interactions. Therefore, we cannot assume that ASVs are superior taxonomic, ecological or evolutionary units than OTUs. This is exactly one of the conclusions reached in this recent paper: <https://doi.org/10.1038/s41467-019-13036-1>. In fact, there are couple of examples showing that an ASV approach does not allow a more precise taxonomic classification than an OTU approach. In a recent study, authors compared skin microbiome and found that ASV and OTU-based approaches were comparable to each other (<https://doi.org/10.1099/jmm.0.001256>). Similarly, this has been discussed in these two papers: <https://doi.org/10.1038/sdata.2019.7>, <https://doi.org/10.1038/s41467-019-13036-1>.

In conclusion, ASV approaches seem not to improve the identification of general patterns of phylogenetically related organisms. Moreover, one rarely expects to find identical bacterial clones across samples because of evolution, drift and local adaptation. Instead one would expect to find distinct but highly similar and phylogenetically closely related organisms. This is why people tend to cluster sequences, or report diversity at higher taxonomic level, e.g. genus or family level, despite the fact that ASV approaches have been used.

To confirm that our results were not biased by using an OTU-approach, we have re-analyzed the amplicon sequencing data with DADA2. In short, the ASV results are very similar to the OTU results. Most importantly, we find the exact same genera predominating the communities, and the unsupervised clustering results in four `pneumotypes' with the same characteristics as obtained by the OTU approach. Some of the most important results are shown below. We have also updated the publicly accessible github repository with an additional file called

"Das_et_al_2020_analysis_DADA2.md", which is a step-by-step markdown of our DADA2 pipeline-based approach:

https://github.com/sudu87/Microbial-ecology-of-the-transplanted-human-lung/blob/master/Das_et_al_2020_analysis_pipeline3_DADA2.md

Given the similarity of the results, we don't see any added value of replacing our OTU-based analysis with the ASVs-based analysis in the manuscript. The fact that two independent approaches give highly similar results corroborates our initial findings.

Detailed results of the comparison between the ASV- and OTU-based analyses are as follows:

1. *Prevalence and abundance of the predominant community members:* Both analyses showed that the most abundant phyla in the BALF samples are *Bacteroidetes* and *Firmicutes*, followed by *Proteobacteria* and *Actinobacteria* (**Figure R1a, b**). Incidence analysis based on ASVs or OTUs identified the same bacteria, *i.e.* *Prevotella 7*, *Streptococcus*, *Veillonella*, *Neisseria*, *Alloprevotella*, *Porphyromonas* (**Figure R1c, d**), as the most prevalent ones. In addition, only ASV_12, corresponding to *Staphylococcus*, was specifically introduced as a result of the ASV-based analysis.

On the contrary, when comparing bacterial prevalence across samples between the analysis based on OTUs and that based on ASVs, we observed a remarkable difference. ASVs reached a maximum incidence of only 25% (**Figure R10c, d**). However, it is important to note that the most prevalent genera from the OTU analysis were further divided into ASVs, thereby resulting in significant reduction of the prevalence of individual ASVs, *e.g.* *Prevotella 7* was represented by 7 ASVs instead of 2 OTUs, (Figure R1c, d).

Figure R1

Figure R1. Abundance and prevalence analysis of taxa from OTU-based and ASV-based analyses. **a,b**, Boxplot showing relative abundance of major phyla obtained from both OTU- and ASV-based analyses. **c,d**, Dot plots showing incidence relative to all BALF samples in the cohort (% , y-axis)

calculated for each OTU or ASV (x-axis). Colored points show genera and OTU IDs or ASV IDs of the taxa present in (c) $\geq 50\%$ or (d) $\geq 5\%$ of BALF samples (red dotted line).

2. *Clustering of samples into pneumotypes*: Using the same unsupervised machine learning algorithm, the samples clustered into four different partitions around medoids (PAMs), independent of the sequence analysis approach, i.e. ASV- vs OTU-based analysis (**Figure R2a, b**).

The heatmap in Figure 2c shows how many samples were assigned to the same PAM/pneumotype across the two analyses. Out of 115 samples assigned to the Pneumotype_{Balanced} (PAM1) in the OTU analysis, 99 were assigned to the same PAM in the ASV analysis. A bit more variation was found across the other three PAMs/pneumotypes. Yet, the largest proportion of samples assigned to PAM2, PAM3, and PAM4 in the OTU analysis were also assigned to distinct PAMs in the ASV analysis. These results highlight that both analyses find similar samples assigned to the four different PAM/pneumotypes. Most importantly, the pneumotype which we associated with a 'healthy' lung state (balanced) showed fewest variation between the two analyses, while the assignment of samples to pneumotypes with disturbed microbiota profiles were sometimes assigned differently.

Figure R2

Figure R2. a, b, Principal component analysis shows Partition around medoids (PAMs) based on OTUs and ASVs respectively, generated by k-medoid-based unsupervised machine learning using Bray-Curtis dissimilarity (occurrence and abundance). **c**, Heatmap of a similarity matrix showing the distribution of pneumotypes (PAMs) in terms of number of samples, based on ASVs (vertical axis) across PAMs based on OTUs (horizontal axis).

3. *Comparison of the clusters obtained on the basis of OTU or ASV analysis*: The four clusters (PAMs) obtained by analysis based on OTUs or ASVs showed similar characteristics in terms of bacterial composition, as indicated in **Figure R3** to **Figure R6**.

Figure R3 pam1- Balanced pneumotype

Figure R3. Comparison of the most abundant OTUs/ASVs present in PAM1 – PneumotypeBalanced – based on the two analyses. The genera *Prevotella 7*, *Streptococcus* and *Veillonella* were the most abundant across all samples (median abundance) in analyses based on **a**) OTUs and **b**) ASVs. In addition, the main genera enriched in terms of average abundance in PneumotypeBalanced (e.g. *Prevotella*, *Alloprevotella*, *Neisseria*, *Gemella*, *Granulicatella*, *Actinomyces*, *Rothia*), according to analyses based on OTUs, also emerged from analyses based on ASVs (Figure 3h and Supplementary Table 4).

Figure R4 pam2- Staphylococcus pneumotype

Figure R4. Comparison of the most abundant OTUs/ASVs present in PAM2 - Pneumotype*Staphylococcus*. PAM2 shows a lower correlation when comparing analyses based on ASVs to those based on OTUs. This is mainly due to the higher representation of *Corynebacterium* in the OTU-based analysis, and *Streptococcus* in the ASV-based analysis. However, in both analyses, *Staphylococcus* is one of the top two genera represented.

Figure R5 pam3- *Microbiota-depleted* pneumotype

negative controls

Figure R5. Comparison of the most abundant OTUs/ASVs present in PAM3 - Pneumotypemd. **a, b**, PAM3, based on the analysis of OTUs or ASVs. This PAM is characterised by the lowest biomass among the four pneumotypes, as shown in our manuscript, hence its designation as "microbiota-depleted". (Figure 3g, i). As we believe that many of the identified taxa resulted from over-sequencing of rare or transient species, or sequencing artefacts, we don't think that the comparison makes too much sense. However, we do find similar taxa in both analyses, predominantly the genus *Flavobacterium*, and this taxa is also predominant in our negative control samples with both analyses (**c, d**) (see also **Supplementray Figure 1a**).

Figure R6 pam4- *Pseudomonas* pneumotype

Figure R6. a, b, Comparison of the most abundant OTU/ASV present in PAM4 - Pneumotype*Pseudomonas* - across the two analyses. PAM4 is highly similar whether identified on the basis of ASVs or OTUs, with *Pseudomonas* predominating in the majority of samples.

Line 297 - "semi quantitative culturing" Why is this not quantitative? I recognize the difficulty with BALF but all of the data is generated from this and it just as semi-quantitative. It would be of great value to compare the CFU recovered to the 16S qPCR for bacterial load. A fundamental difficulty in the lower airways in culture-independent methods is the low turnover of DNA that accumulates and this gets proportionally worse with disease.

Reply: This statement refers to the results shown in Fig. 3j. As part of the routine clinical investigations for diagnostic purposes, a microbiological assessment of the predominant bacteria in each BALF and bronchial aspirate (BA) sample was carried out. To this end, samples were cultured on selective media for discriminating between major pathogens (*Pseudomonas aeruginosa*, *Staphylococcus aureus*), and the commensal oropharyngeal flora. The results obtained from this analysis were only qualitative (culture positive or negative) and not quantitative (no dilution series performed). In Fig 3j, we use these results to report the percentage of BALF samples that were identified to have bacteria associated with each of the pneumotypes. We agree that the term semi-quantitative is misleading in this context and replaced it with qualitative in line 305 of the revised manuscript.

line 537 - "The high variability in community composition and bacterial load between BALF samples may suggest that the human lung microbiota is highly erratic." This is a very unusual statement- and I am not sure what the authors are implying.

Reply: We agree. The term was changed to "highly variable".

A major claim is that this is one of only a few studies to apply culturing to the lung microbiome. (line 83 and elsewhere e.g line 234). This ignores a large body of work from chronic airway disease where this has been done quite extensively (and more comprehensively than reported here). While less has been published on the healthy lung microbiota, I would point out that the transplanted lung is not equivalent to a healthy lung and caution should be taken in interpreting it as such.

Reply: We thank the reviewer for pointing this out and we apologize for not making it clear. Firstly, we have changed the sentence in line 85-88, as follows to acknowledge the culturing approaches carried out in previous studies:

"Thirdly, while several studies have isolated viable bacteria from lung samples^{6,13,14}, a bacterial collection that can serve as a public resource has not been established, and little is known about the physiology and growth characteristics of lung isolates."

Please note that the idea of our in-house culture collection is to offer a resource for other scientists in the field and to use it to further elucidate the phylogeny and physiological properties of lung bacterial isolates: https://github.com/sudu87/Microbial-ecology-of-the-transplanted-human-lung/blob/master/LuMiCol_part1.csv

Secondly, we do not claim that the transplanted lung is equivalent to the healthy lung. However, our balanced pneumotype is similar to the microbiota reported for the healthy lung (Segal et al. 2013 and 2016). This is what we point out on line 509, where we conclude:

"Our analyses show that the human lung microbiota post-transplant predominantly consists of oropharyngeal taxa similar to the microbiota of healthy lungs^{5,6,26}. Hence, the presented results are not only relevant in the context of lung transplantation, but also provide general insights into the microbial ecology of the lower respiratory tract."

It would have been informative to have more longitudinal analysis. Although this is done in an overview at the pneumotypes in Fig 7, this is a missed opportunity as there is regular sampling at the same time intervals in over half the samples (e.g. 3,6,12,24 months)

We agree with the reviewer that the longitudinal approach could have been further extended by testing for example for the conservation of ASVs/OTUs over time in a given sample. However, the major focus of our study was to characterize the communities on the level of the identified pneumotypes and in this respect our longitudinal analyses shows that there is no significant difference in the distribution of pneumotypes between the time windows (**Figure 7**), but that Pneumotype_{Balanced} shows the greatest stability and that the first two time windows show greater resiliences than the others. Given the already quite substantial results sections and the fact that additional analyses would not necessarily be in line with the pneumotype focus of our manuscript, we would not like to extend this part further at this point

There was no discussion of pre-transplant patient infections. The upper respiratory tract is a reservoir for lower airway pathogens and most transplant patients in chronic airway disease (particularly CF and bronchiectasis) that get re-infected are by the same STRAIN they were previously infected by. Do the pneumotypes defined by *Staphylococcus* and *Pseudomonas* respectively represent patients that had these infections prior to transplant?

Reply:

We thank the reviewer for this comment and acknowledge the importance of providing information on the impact of pre-transplant colonisation on post-transplant pneumotypes. The fact that we have no means to determine the Pneumotypes pre-transplant, we tried an indirect way to address this question. For this purpose, we used patients records of upper respiratory bacterial colonisation (bronchial aspirates, similar as shown in **Figure 3j**).

Hence, we have performed a Chi square test for independence to look for an association between these categories in the revised manuscript. We examined exclusively Pneumotype_{Staphylococcus} and Pneumotype_{Pseudomonas}, as the inclusion of Pneumotype_{Balanced}, which had a very low incidence of *Staphylococcus aureus* (Sa) or *Pseudomonas aeruginosa* (Pa) and may bias the association test. Based on the patients records of pre-transplant and post-transplant colonization status, as obtained from routine clinical assessment, we made individual contingency tables, whereby "yes/no" was attributed to either colonisation by *S. aureus* (**Table R1a**) or *P. aeruginosa* (**Table R1b**), in the Pneumotype_{Staphylococcus} and Pneumotype_{Pseudomonas} respectively.

Figure R8

Figure R8. a,b, Count comparison of *Staphylococcus aureus* and *Pseudomonas aeruginosa* colonisation pre- (bar colors) and post-transplant (x-axis) in association with Pneumotype_{Staphylococcus} and Pneumotype_{Pseudomonas} respectively.

We found that the distribution of bacterial colonisation was random and that the association between pre- and post-transplant colonisation was not significant, neither for *S. aureus* (**Figure R8a**, $\chi^2 = 0.047$, $p = 0.82$) nor for *P. aeruginosa* (**Figure R8b**, $\chi^2 = 0.2$, $p = 0.65$). To summarize, both the presence of Pneumotype*Staphylococcus* and Pneumotype*Pseudomonas* in the transplanted lung did not necessarily imply pre-transplant colonisation by *S. aureus* and *P. aeruginosa*, respectively. In addition, pre-transplant colonisation by these bacteria did not invariably imply the presence of the corresponding pneumotype after transplantation (**Figure R8a, b**).

Hence, we have added the following lines to make it apparent in the manuscript text (line 317-321) and added a new figure panels as Supplementary Fig 5c, d.

*"This is supported by the fact that for Pneumotype*Staphylococcus* and Pneumotype*Pseudomonas* in the transplanted lung, there was no association between pre- and post-transplant colonization by *S. aureus* (Chi-square test, $p = 0.82$) and *P. aeruginosa* (Chi-square test, $p = 0.65$) (Supplementary Fig 5c, d), respectively, in the upper respiratory tract."*

Minor issues:

line 108-109. The problem with these types of analyses/generalizations is the talking about these at the genera level. While there are large numbers of Streptococcus species that differ dramatically in their pathogenic potential (a Group B strep is hardly equivalent to a *S. salivarius*).

We thank the reviewer for pointing this out. We do not intend to generalize and are well aware of the effect of pathogenic *Streptococci*. Here we refer to our previous study, Reference 25: Bernasconi, E. et al. 2016 Am. J. Respir. Crit. Care Med, which showed the low inflammatory impact of the non-pathogenic *Streptococcus pneumoniae* D39 strain in co-culture *in vitro* with monocyte-derived macrophages. To be more specific, we have modified the sentence in 113-115, as follows:

*"This is in contrast to certain strains of non-pathogenic *Streptococcus pneumoniae*, whose abundance has been linked to low inflammation and tissue repair and remodelling"*

line 239 It is well known that anaerobic growth of *Pseudomonas* required nitrate (or another terminal electron acceptor) this is sufficient nitrate present in chronic airway disease to support anaerobic growth of *Pseudomonas* and there is data to suggest that it does grow under anaerobic conditions.

Here we refer to the culturing in broth (liquid medium), in which we did not see any growth of *Pseudomonas OTU_1* under anaerobic atmosphere (**Figure 2b**). However, in agreement with the reviewer's comment, we saw growth of *Pseudomonas* on semi-solid agar in anaerobic condition (**Figure 2b**, panel 'Semi-solid media' with colored pie charts describing oxygen conditions). To avoid confusion, we have modified the sentence in line 245-247 to explicitly mention "liquid medium". The sentence reads now as follows:

*"The two most predominant opportunistic pathogens in our lung cohort, *P. aeruginosa* (OTU_1) and *S. aureus* (OTU_2), grew best in rich liquid medium in the presence of oxygen (Fig. 2c), although both also grew under anaerobic conditions."*

242 "...these results indicate towards changes physicochemical conditions in the lung that may favor the growth of aerobic bacteria with potentially pathogenic properties." This is a rather broad statement that I do not think is very accurate. These are two opportunistic pathogens that are the most common to infected patients with chronic airway disease, and must be well adapted to do this not just any "aerobic bacteria with potentially pathogenic properties" which is a pretty long list.

Reply: We agree that this statement was too general and should specifically refer to the two opportunistic pathogens. The sentence in line 247-249, has been modified as follows:

"These results indicate that changes in the physicochemical conditions in the lung may favor the growth of these two opportunistic pathogens."

The authors do seem to use strain and species interchangeably and should be cautious not to. e.g. lines 525-527. While most species can be resolved by 16S (not all but most even in the streptococci) just not always with short amplicons, it is exceptional to resolve strains by the 16S gene. There are varying degrees of pathogenicity between strains within a species - it is not clear if that is what they are referring to here or species resolution.

Reply: We have carefully looked through the text and have identified three places where the term 'strain' was misleading and hence was replaced with 'species' for reasons of consistency (Lines 125, 329 and 534 in the clean version of the revised manuscript).

line 535 - numerous studies have demonstrated most of these organisms are readily cultured with very few exceptions in the airways. It could be the choice of culture conditions chosen here which may have limited their success.

Reply: We agree with the reviewer that most bacteria are culturable, and therefore have included the following two sentences in the manuscript (Lines 538-542):

"We acknowledge that the presented culture collection is not exhaustive. We believe that this is most likely due to the high variability of the lung microbiota and the fact that we have cultured a relatively small number of BALF samples, rather than the inability of some community members to grow in vitro under the tested conditions, or their non-viability in the lungs."

line 551-552. Staphylococcus is one of the most common lower respiratory pathogens in chronic airway disease and not uncommon as an acute respiratory pathogen.

We apologize for the misunderstanding. Indeed, *Staphylococcus* is a common lung opportunistic pathogen in both acute and chronic disease. *S. aureus* is also a very common inhabitant of the nasal mucosa, with almost 50% healthy adults colonized in the anterior nares. *Staphylococcus* is highly adaptable and evolves with its host, concealing itself from immune system to cause infection (<https://doi.org/10.1073/pnas.1520255113>). Recently, *Staphylococcus* was also found as one of the first colonizers in neonatal airways ([10.1016/j.chom.2018.10.019](https://doi.org/10.1016/j.chom.2018.10.019)). Hence, here we tried to make a link to its colonization of human airways in general not only its pathogenic potential of humans. We have therefore modified the sentence in line 558-560, as follows:

"Staphylococcus, the major community member of Pneumotype_{Staphylococcus}, is a frequent colonizer of the healthy human nasal mucosa capable of evolving on its host and switching to being infectious. It has also been shown to dominate in neonatal lower airways, indicating potential early adaptation to human lung."

In the culture conditions table, 5%CO₂ is described as micro-aerobic. It is not.

Reply: We apologize for this misunderstanding. This has been modified to "in presence of 5% CO₂" in main text and methods. In Supplementary Data 4 i.e. culture conditions table, the term "micro-aerobic" has been removed.

As a final point, which will know doubt be contentious but highly relevant (and I will not categorize as a major or minor issue but just a comment). I do have a fundamental problem with the concept of describing the microbiota of chronic airway disease as a dysbiosis. Many in the field (and the best data) argues that the healthy lung is transiently colonized (with very little growth) with a very low biomass and cleared. This transient colonization is no doubt an important immune modulator of the lung. But this is very different that the chronic colonization with orders of magnitude higher bacterial burden that is actively growing in the lower airways. This is not really the dysbiosis as described at other body sites which have resident growing communities of microbiota (not that the term dysbiosis is fraught with problems of its own...).

Reply: We thank the reviewer for raising this interesting point, which is at a crossroad between the analysis of microbial communities, semantics, and even philosophy. Although there seems to be little debate that dysbiosis (as long as the term is well defined in a given clinical context) can be identified in supraglottic anatomical sites, where bacteria are abundant and relatively stable even in the healthy state, translation to the lower respiratory tract can indeed be questioned. In fact, our view is essentially in line with the reviewer's position. Therefore, to avoid any controversial statements, we have removed in the revised manuscript the 2 initial references to lung dysbiosis, which now read as follows:

Line 70-71: "Shifts in microbial community composition, together with a decrease in bacterial diversity, have been associated with various respiratory diseases..."

Line 634-636: "Overall, we propose that the four pneumotypes seem to follow the "Anna Karenina principle", where healthy communities vary little around a stable state, while perturbed communities are much more variable with unstable states."

Reviewer #3 (Remarks to the Author):

The following comments are focused (as requested by the editor) on the data analysis and the ML aspects.

The main problem of this work is rarefaction. Looking at the text and code it seems that no rarefaction was performed on the microbiome data.

While rarefying is not an optimal normalization method, as it reduces the statistical power, it is essential to reduce the artifacts due to the differences in library sizes across samples (doi:10.1186/s40168-017-0237-y). Authors should therefore apply rarefaction (whithout replacements) before alpha/beta, statistical and machine learning analyses in order to obtain meaningful results.

Reply: Thank you for pointing this out. We apologize to not have stated this in the method section. The data was in fact rarefied to the sample with the minimum number of reads. We have edited the method text in 'Data processing and OTU picking' and now explicitly state that the data was rarefied on line 844-845:

" After quality filtering and rarefying the data to the sample with the lowest number of reads , we ended up with 7164 OTUs at 97% identity in 16S rRNA gene"

Other points

Data processing and OTU picking:

- Line 818: What did you mean by : "... as this increases the risk of obtaining spurious taxa, we removed OTUs with ambiguous taxa i.e. NA ..."? Did you remove any unclassified OTU? At which taxonomic level? Please explain this point better.

Reply: We apologize that this was not clear enough from our text. OTUs that were not assigned to the taxonomy at phylum level were removed. In addition, we applied a filter to improve the analysis, by using the following code:

We filtered OTUs by using 'subset_taxa' function of phyloseq R package that were either unassigned or assigned as "NA" by mapping and SILVA taxonomy. Source: [https://github.com/sudu87/Microbial-ecology-of-the-transplanted-human-lung/blob/master/Das et al 2020 analysis pipeline 2.md](https://github.com/sudu87/Microbial-ecology-of-the-transplanted-human-lung/blob/master/Das%20et%20al%202020%20analysis%20pipeline%202.md)

This can be found in the pipeline mentioned above at step no. 19.

We have therefore modified the following text in the 'Methods' section 'Data processing and OTU picking' in line 835-841:

"Before proceeding further, we removed OTUs with ambiguous taxa i.e. Phylum= "NA" and any samples that had less than 10⁴ reads and no information on bacterial abundance. In conventional practice, samples with low biomass (detected by 16S qPCR) are excluded but we elected to retain them if sequencing was successful (≥10⁴ reads). However, in these low biomass samples the risk of obtaining spurious taxa is higher, requiring comparisons with negative controls, which included Bronchoscope pre-wash, DNA extraction reagents and no-template PCR reaction."

- Line 825: "... After quality filtering and normalization, ..." Which normalization?

Reply: Please, see comment above. We apologize for not explaining this properly. The data was normalized by rarefying to the sample with the lowest number of reads, as indicated at step 21 in the pipeline below. We have updated the code on https://github.com/sudu87/Microbial-ecology-of-the-transplanted-human-lung/blob/master/Das_et_al_2020_analysis_pipeline_2.md

Extraction of OTU-isolate match pairs:

- This part is difficult to understand. Please rewrite it in a professional and detailed form.

Reply: The corresponding method section at line 849 has been rewritten, and reads as follows :

"In order to assign the cultured bacteria to OTUs (i.e. OTU-isolate match pairs), we aligned the high quality 16S rRNA gene sequences obtained by Sanger sequencing from 215 LuMiCol isolates (phred score: Q30 > 90%) to the dereplicated Illumina sequencing 16S rRNA amplicons using MAFFT alignment tool in Geneious Prime software (Geneious, New Zealand). The LuMiCol sequences were then trimmed to same size as the 16S rRNA amplicons and included in the community analysis as new sample "LuMiCol" after the dereplication step and before the last mapping step (--usearch global; Step 13- Das_et_al_2020_analysis_pipeline_1.md; <https://github.com/sudu87/Microbial-ecology-of-the-transplanted-human-lung>) for OTU and taxonomic assignment (See also "Data processing and OTU picking"). OTUs containing Sanger sequence reads from the sample "LumiCol" were identified and flagged as OTU-isolate matching pairs."

Unsupervised learning for Pneumotype discovery:

- Genocrunch parameters and detailed output are missing.

Reply: We have downloaded the genocrunch parameters and attached them as Supplementary Data 7 (zip file). The file bundle consists of extensive input parameters; json and text files and output analyses and Silhouette plots. We have also added two files consisting of input and output parameters, 'genocrunch_input_parameters_genocrunch.json' and 'genocrunch_output_parameters_genocrunch.json', and related description on our GitHub repository (<https://github.com/sudu87/Microbial-ecology-of-the-transplanted-human-lung>).

We have updated the methods at line 935 accordingly which now reads as follows:

"Pneumotypes were obtained by running k-medoid-based unsupervised machine learning on a Bray-Curtis dissimilarity matrix (binary = FALSE) using Genocrunch (see Section "Statistical analysis and software"). The program utilizes the pamk function of the R package 'fpc' version 2.1.10 to cluster samples while optimizing the number of clusters based on the average silhouette width^{72,73}. Input and output parameters used in Genocrunch were provided in the corresponding json files, which are available on the GitHub repository (<https://github.com/sudu87/Microbial-ecology-of-the-transplanted-human-lung>)."

Random Forest:

- It is not clear how the models were optimized for best accuracy. Please write the section as detailed and clear as possible.

Reply: We thank the reviewer for pointing this out and apologize for not having provided sufficient details about the optimisation of the models we used. Below we have outlined all the different steps for the referee in brief and have updated the existing pipeline with detailed explanation in https://github.com/sudu87/Microbial-ecology-of-the-transplanted-human-lung/blob/master/Das_et_al_2020_analysis_pipeline_2.md in Steps 56-58.

We have updated the Methods section with a short description and referred to our GitHub Repository for full explanation of the optimization process.

Specifically, the section 'Statistical analysis and Software used' was updated with the following text, in lines 664:

"For optimization of random forest parameters, the caret package was used in R."

We also added in Methods, in section 'Machine learning and statistical modelling', a new subsection called 'Optimization of Random Forest parameters' with the following text (lines 945-982):

" We used Random Forest to predict i) the pneumotype and ii) the bacterial and viral counts based on host gene expression data. The median normalized expression levels of the 31 analyzed host genes presented in Fig. 4a from all 234 subject samples were used as predictors. For predicting pneumotypes, we used a classification model and for predicting bacterial or viral loads, we used a regression model.

While machine learning algorithms typically use a training and a prediction set, Random Forest allows for cross-validation, i.e. random shuffling of existing data, allowing predictions to be made without the need for an external prediction data set. Such cross-validation was first performed ten times and then iterated three times. We optimized the models using the 'caret' package in R, by tuning the two most important parameters: i) the mtry (i.e. splits per try - number of randomly selected predictor genes to be sampled at each iteration), and ii) the ntrees (i.e. number of decision trees), using a specific mtry at each iteration. Full details on the entire optimization process and the code can be found on the GitHub repository ([https://github.com/sudu87/Microbial-ecology-of-the-transplanted-human-lung/blob/master/Das et al 2020 analysis pipeline 2.md](https://github.com/sudu87/Microbial-ecology-of-the-transplanted-human-lung/blob/master/Das%20et%20al%202020%20analysis%20pipeline%202.md)) from Steps 56-58. In short, to optimize mtry in case of pneumotype prediction, we used the 'grid search' function (10-fold cross-validation, with three iterations of the whole process). For regression models, we used the 'random search' function, whose 'tuneLength' argument randomly specifies the number of mtry values, which we set to the maximum of 30. We found that mtry of 5, 21 and 10 gave the best accuracy results for pneumotype, bacterial counts, and viral count prediction respectively. Using these values, we optimized the number of decision trees (ntrees) needed for good predictions, by running the 'grid search' or 'random search' functions within an incremental loop to vary the ntrees from 500 to 5000 for both the classification and regression models. For the classification model predicting pneumotypes, we found little difference in accuracy and sensitivity percentages. Hence, we kept the minimum number of trees i.e 500.

For regression models predicting bacterial and viral counts, instead of accuracy or sensitivity percentages, the results were assessed by low Root Mean Squared Error (RMSE) and high regression value i.e. R2. Similarly, we observed little difference in RMSE and R2 with the series of ntrees tested, and used ntree= 1000 for bacterial number prediction and ntree= 500 for viral count prediction, which were used by the algorithm as defaults."

- Line 935: Please explain how importance scores are computed.

Reply: Importance scores were calculated by Boruta package, as previously described in "Kursa, M., & Rudnicki, W. (2010). Feature Selection with the Boruta Package. *Journal of Statistical Software*, 36(11), 1 - 13. doi:<http://dx.doi.org/10.18637/jss.v036.i11>"

We have updated the approach of Boruta in the Materials and Methods section (line 976-993). It reads as follows:

"After every analysis, random forest provides results in terms of error rate (out-of-box error) and table for the classification predictions (Supplementary Table 5) and percentage of variance explained for regression predictions (Fig. 4i and 5f). This indicates the importance of each feature, i.e. gene predictor, which is defined by the increase in accuracy (in classification models) and variances (in regression model) at every step in a decision tree, while using a particular feature amongst others. 'Boruta' algorithm creates several copies of the data where it randomly shuffles each feature. These

permuted features (shadow features) are attached to existing data, on which the random forest function is then applied. At every iteration during a decision tree, the algorithm compares the original features to a threshold, which is created by recording the highest importance within the shadow features (i.e. shadowMax). Additional thresholds are also created, shadowMin, i.e. the minimum importance within the shadow features, and shadowMean, i.e. the mean importance within the shadow features. The features that pass this threshold i.e. having higher Z score than the shadowMax are assigned as "Confirmed" while other features are "Rejected". However, since a random forest algorithm runs for a limited number of steps, features remain that pass the threshold but cannot be confirmed and are hence assigned as "Tentative". This can be handled by using the function 'TentativeRoughFix' that fills the missing decisions by comparing the median Z score of a feature to the median of the most important shadow feature (i.e. shadowMax).

- Lines 939-941: "Gene predictors from random forest analysis with importance scores more than 10 were further fitted into additive linear model with either bacterial or viral copy numbers ..."
Please explain why this makes sense since the RF classifier is strongly non linear.

Reply: We acknowledge that Random Forest is a non-linear statistical approach. While it helped us in identifying biomarkers for bacterial and viral number predictions, it did not inform us about the direction (negative or positive) of change in gene expression. Therefore we confirmed/complemented the top random forest hits by the conventional linear statistical approach. For example, the biomarkers Platelet-derived growth factor D (PDGFD) and Interferon predicted by RF, was found by the linear statistics approach to have the highest expression in a context of microbiota depletion, i.e. in association with PneumotypemD, from which hypotheses can be formulated on how this gene is involved in microbiota-host crosstalk.

Please explain why you have taken predictors with importance scores more than 10.

We acknowledge that this threshold is essentially arbitrary and we do not conclude that the rest of the "Confirmed" predictor genes are insignificant. To avoid to expand the methods too much, we have decided to limit the presentation of linear statistics to the top predictors. We opted for a threshold of 10 because in two of the three analyses, there was a gap between genes with a score above or below this cut-off. In the case of viral count predictors (**Figure 5f**) we only found four "Confirmed" predictor genes, which we all picked for validation with linear statistics.

Code availability:

- The code should be uploaded to public code repositories, like Github or Bitbucket.

Reply: The code was submitted to GitHub (see link below) at the time of the initial submission, and is also indicated in the Methods section on lines 668.

<https://github.com/sudu87/Microbial-ecology-of-the-transplanted-human-lung>

REVIEWERS' COMMENTS

Reviewer #2 (Remarks to the Author):

The authors have provided a comprehensive response to the reviewer comments and I am over all satisfied with the responses.

I commend the authors for a comprehensive comparison of the OTU and ASV approaches to analyzing their data, and while I agree the data are qualitatively very similar, there are differences which I respectfully disagree with the authors that these are not trivial differences . Furthermore, ASVs are more reproducible and easier to compare across datasets, and consequently being required from some journals now. I also recognize that there is no one "right way" to analyze microbiome data and accept the authors decision to stay with OTU clustering if that is their choice.

The inclusion of more details around the methodology is appreciated.

Reviewer #3 (Remarks to the Author):

The issues raised in the review have been satisfactorily addressed.

REVIEWERS' COMMENTS

Reviewer #2 (Remarks to the Author):

The authors have provided a comprehensive response to the reviewer comments and I am over all satisfied with the responses.

Reply: We are pleased that Reviewer 2 appreciated our efforts to respond to his/her pertinent comments. In doing so, we feel that the manuscript has further improved in quality.

I commend the authors for a comprehensive comparison of the OTU and ASV approaches to analyzing their data, and while I agree the data are qualitatively very similar, there are differences which I respectively disagree with the authors that these are not trivial differences . Furthermore, ASVs are more reproducible and easier to compare across datasets, and consequently being required from some journals now. I also recognize that there is no one "right way" to analyze microbiome data and accept the authors decision to stay with OTU clustering if that is their choice.

The inclusion of more details around the methodology is appreciated.

Reply: We are pleased that Reviewer 2 acknowledges the qualitative similarity between the data sets obtained by an ASV-based approach compared to OTUs. Although there is room for a thematic discussion on the significance of the observed differences, it was important for us to show that the two approaches converge towards the identification of similar pneumotypes after transplantation, which ultimately reinforces the existence of these different community profiles. We are also pleased that the inclusion of more details on the methodology has been helpful.

Reviewer #3 (Remarks to the Author):

The issues raised in the review have been satisfactorily addressed.

Reply: We are pleased that our responses to the pertinent matters raised by Reviewer 3 were found satisfactory.